# Decoding Neurodegeneration: A Review of Molecular Mechanisms and Therapeutic Advances in Alzheimer’s, Parkinson’s, and ALS

**DOI:** 10.3390/ijms252312613

**Published:** 2024-11-24

**Authors:** Corneliu Toader, Calin Petru Tataru, Octavian Munteanu, Matei Serban, Razvan-Adrian Covache-Busuioc, Alexandru Vlad Ciurea, Mihaly Enyedi

**Affiliations:** 1Department of Neurosurgery, “Carol Davila” University of Medicine and Pharmacy, 050474 Bucharest, Romania; corneliu.toader@umfcd.ro (C.T.); matei.serban2021@stud.umfcd.ro (M.S.); razvan-adrian.covache-busuioc0720@stud.umfcd.ro (R.-A.C.-B.); prof.avciurea@gmail.com (A.V.C.); 2Department of Vascular Neurosurgery, National Institute of Neurology and Neurovascular Diseases, 077160 Bucharest, Romania; 3Ophthalmology Department, “Carol Davila” University of Medicine and Pharmacy, 020021 Bucharest, Romania; 4Department of Anatomy, “Carol Davila” University of Medicine and Pharmacy, 050474 Bucharest, Romania; mihaly.enyedi@umfcd.ro; 5Neurosurgery Department, Sanador Clinical Hospital, 010991 Bucharest, Romania; 6Medical Section within the Romanian Academy, 010071 Bucharest, Romania

**Keywords:** Parkinson’s disease (PD), Alzheimer’s disease (AD), Huntington’s disease (HD), protein misfolding, neuroinflammation, mitochondrial dysfunction, gene therapy, precision medicine, pharmacogenomics

## Abstract

Neurodegenerative diseases, such as Alzheimer’s, Parkinson’s, ALS, and Huntington’s, remain formidable challenges in medicine, with their relentless progression and limited therapeutic options. These diseases arise from a web of molecular disturbances—misfolded proteins, chronic neuroinflammation, mitochondrial dysfunction, and genetic mutations—that slowly dismantle neuronal integrity. Yet, recent scientific breakthroughs are opening new paths to intervene in these once-intractable conditions. This review synthesizes the latest insights into the underlying molecular dynamics of neurodegeneration, revealing how intertwined pathways drive the course of these diseases. With an eye on the most promising advances, we explore innovative therapies emerging from cutting-edge research: nanotechnology-based drug delivery systems capable of navigating the blood–brain barrier, gene-editing tools like CRISPR designed to correct harmful genetic variants, and stem cell strategies that not only replace lost neurons but foster neuroprotective environments. Pharmacogenomics is reshaping treatment personalization, enabling tailored therapies that align with individual genetic profiles, while molecular diagnostics and biomarkers are ushering in an era of early, precise disease detection. Furthermore, novel perspectives on the gut–brain axis are sparking interest as mounting evidence suggests that microbiome modulation may play a role in reducing neuroinflammatory responses linked to neurodegenerative progression. Taken together, these advances signal a shift toward a comprehensive, personalized approach that could transform neurodegenerative care. By integrating molecular insights and innovative therapeutic techniques, this review offers a forward-looking perspective on a future where treatments aim not just to manage symptoms but to fundamentally alter disease progression, presenting renewed hope for improved patient outcomes.

## 1. Introduction

### 1.1. Overview of Neurodegenerative Diseases

Neurodegenerative diseases, such as Alzheimer’s disease (AD), Parkinson’s disease (PD), amyotrophic lateral sclerosis (ALS), and Huntington’s disease (HD), present a unique scientific challenge due to their complexity and the devastating effects they have on cognitive and motor function [1]. Though each disease has its distinct characteristics, they share underlying molecular mechanisms such as oxidative stress and protein misfolding. For instance, Alzheimer’s disease and Parkinson’s disease are characterized by the accumulation of misfolded proteins like amyloid-beta and alpha-synuclein, respectively. These misfolded proteins propagate toxicity and disrupt cellular function, highlighting a shared pathological hallmark [2]. These diseases highlight an urgent need for deeper understanding as they continue to affect millions worldwide, straining healthcare systems and impacting families profoundly. For example, Parkinson’s disease is increasingly prevalent, affecting over 10 million people worldwide, and Alzheimer’s disease alone contributes to an economic burden exceeding $1 trillion annually due to healthcare costs and lost productivity [3].

In Alzheimer’s disease, a major culprit is the amyloid-beta (Aβ) peptide, a fragment derived from the amyloid precursor protein (APP) [4]. When APP is cleaved by enzymes known as beta- and gamma-secretases, Aβ fragments are produced, which are prone to clumping together outside cells [5]. These clumps, or plaques, have long been considered a hallmark of AD, but recent research has taken a closer look at the soluble forms of Aβ, suggesting that these smaller aggregates might be the real neurotoxic agents [6]. Soluble Aβ oligomers can interact directly with cell membranes, disrupt ion channels, and trigger internal signaling pathways that interfere with synaptic function, eventually leading to neuronal death [7]. Adding to this, tau protein—a stabilizing component of the neuronal skeleton—undergoes hyperphosphorylation in AD, transforming into tangled filaments within neurons [8]. These tau tangles not only weaken neuronal structure but also appear to spread across brain regions in a manner that mirrors the disease’s clinical progression [9].

In Parkinson’s disease, the loss of dopamine-producing neurons in the substantia nigra region of the brain leads to the characteristic motor symptoms of tremors and rigidity [10]. At the heart of this loss lies alpha-synuclein, a protein that, under certain conditions, misfolds and accumulates inside neurons, forming structures called Lewy bodies. Studies have revealed that alpha-synuclein has an intrinsic ability to change shape, creating beta-sheet-rich structures that can seed aggregation in nearby cells, a behavior often described as “prion-like” [11]. This propagation potentially explains how Parkinson’s pathology progresses through interconnected brain regions. Alongside protein misfolding, Parkinson’s pathology includes significant mitochondrial dysfunction [12]. Key genes, such as PINK1 and Parkin, which are involved in mitochondrial quality control, become dysfunctional, impairing the cells’ ability to clear damaged mitochondria [13]. This disruption leads to increased oxidative stress, further endangering the survival of dopamine-producing neurons [14].

In ALS, nearly all cases show an accumulation of a protein known as TDP-43 within motor neurons [15]. Normally, TDP-43 is found in the nucleus where it plays a vital role in RNA processing [16]. However, in ALS, it becomes abnormally modified and forms toxic aggregates in the cytoplasm [17]. This mislocalization disrupts essential cellular functions, creating stress that leads to motor neuron degeneration [18]. Genetic studies have also identified mutations in *SOD1*, a gene associated with familial ALS that contribute to mitochondrial dysfunction and oxidative damage, core features of ALS pathology [19]. Mitochondrial dysfunction in neurodegenerative diseases stems from disrupted oxidative phosphorylation, which reduces ATP production and generates excess reactive oxygen species (ROS). These ROS damage lipids, proteins, and mitochondrial DNA (mtDNA), perpetuating a cycle of oxidative stress. Impaired dynamics, characterized by disrupted fusion and fission processes, further fragment mitochondria, compromising their transport and energy distribution in axons and synaptic terminals [20].

Defective mitophagy exacerbates the problem, as seen in Parkinson’s disease, where impaired PINK1-Parkin recruitment prevents the clearance of dysfunctional mitochondria. Similarly, in ALS, mutations in SOD1 amplify oxidative damage and disrupt calcium buffering. Suppressed mitochondrial biogenesis, regulated by the PGC-1α/NRF1/TFAM axis, limits the replacement of damaged organelles, deepening the energy crisis. Together, these processes create a critical vulnerability in neurons, driving excitotoxicity, synaptic failure, and cell death [21].

Advances in cell modeling, particularly using stem cell-derived neurons, have enabled scientists to observe how ALS-related proteins interfere with cellular transport, RNA processing, and stress responses, all of which contribute to the decline of motor neurons [22].

Huntington’s disease has its own distinctive molecular feature: an expansion of CAG repeats in the *HTT* gene, leading to an extended polyglutamine tract in the huntingtin protein [23]. This mutant huntingtin (mHTT) misfolds and accumulates in neurons, primarily within the striatum, a region central to motor function and coordination. These aggregates disrupt normal cellular processes by binding to essential transcription factors, leading to widespread changes in gene expression [24]. Moreover, mHTT is known to interfere with mitochondrial function, impairing energy production and increasing oxidative stress, compounding the damage to vulnerable neurons [25].

Common Molecular Hallmarks

Although these diseases stem from different genetic and molecular origins, they converge on shared pathways that reflect common mechanisms of neurodegeneration. Protein misfolding and aggregation, for example, are central to each disease’s progression [26]. In Parkinson’s disease, α-synuclein exhibits prion-like behavior, propagating through interconnected brain regions by inducing misfolding in neighboring neurons. Similarly, in Alzheimer’s disease, tau protein spreads via synaptic connections, highlighting a prion-like mechanism for disease progression [27].

Mitochondrial dysfunction and oxidative stress also play crucial roles as neurons rely heavily on energy and are particularly vulnerable to mitochondrial impairments [28]. When mitochondria malfunction, they produce ROS, leading to lipid peroxidation, protein oxidation, and DNA damage—each a destructive force that accelerates cellular aging and death. Neuroinflammation plays a dual role in neurodegeneration. While acute neuroinflammation serves protective functions by clearing debris and repairing tissue, chronic activation of microglia and astrocytes releases pro-inflammatory cytokines that exacerbate neuronal damage. This context-dependent role underscores the complexity of targeting neuroinflammation in therapeutic strategies [29].

This convergence of molecular pathways across neurodegenerative diseases emphasizes the need for a holistic, systems-based approach to understanding these interconnected mechanisms. Increasingly, researchers are looking for biomarkers and therapeutic targets that cut across these shared pathways, with the aim of developing treatments that could potentially benefit multiple neurodegenerative diseases [30].

### 1.2. The Importance of Molecular Understanding

The molecular exploration of neurodegenerative diseases has revolutionized the field, shifting our perspective from broad disease categories to precise, mechanistic understanding [31]. By dissecting the fundamental cellular and molecular pathways involved in diseases like AD, PD, ALS, and HD, scientists are uncovering how these intricate biological processes converge, sparking innovations in diagnostics and therapeutic strategies [32].

One of the pivotal advancements in recent years has been the identification and development of biomarkers. Biomarkers for AD, for instance, now include specific protein signatures detectable in cerebrospinal fluid (CSF) and blood, providing insight into disease progression long before symptoms arise [33]. Elevated levels of Aβ and phosphorylated tau—key pathological proteins in AD—are now measurable through advanced techniques such as ultrasensitive immunoassays and liquid chromatography–mass spectrometry [34]. These biomarkers provide a critical glimpse into the “silent” phases of neurodegeneration, often detectable as early as 20 years before cognitive symptoms manifest and, thus, hold enormous potential for early intervention and tracking disease progression in clinical settings [35].

Moreover, molecular insights have clarified the interwoven pathways that drive neurodegeneration, revealing how initial cellular disturbances propagate across interconnected networks [36]. When proteins like tau in AD or alpha-synuclein in PD misfold and aggregate, they not only form toxic intracellular clumps but also disrupt surrounding cellular systems [37]. These aggregates place substantial stress on mitochondria, the cell’s energy factories, impairing their function and leading to the generation of ROS. This oxidative stress damages essential cellular components like DNA, proteins, and lipids, creating a vicious cycle where protein misfolding and cellular damage fuel each other [38]. This cascade, involving both mitochondrial dysfunction and oxidative stress, is a theme across many neurodegenerative diseases and has underscored the importance of developing multi-target therapies that can simultaneously stabilize protein structures and protect mitochondrial health [39].

Targeted therapeutics represent another area transformed by molecular research. Traditional treatments often aimed to alleviate symptoms but did not address the disease’s root molecular causes. Now, therapies are being designed with unprecedented precision [40]. For example, small molecule inhibitors have been developed to specifically prevent amyloid and tau aggregation in AD and alpha-synuclein aggregation in PD [41]. These inhibitors bind selectively to misfolded proteins, blocking the formation of toxic aggregates and, in preclinical studies, have shown potential to reduce plaque formation and improve neuronal health [42]. Another promising avenue involves molecular chaperones—proteins that assist in proper folding and prevent misfolding—showing efficacy in reducing neurotoxic aggregates in experimental models of diseases like HD and ALS [43]. Such approaches mark a significant departure from earlier symptomatic treatments as they focus on halting or even reversing molecular dysfunction at its source [44].

Gene-editing technologies, particularly CRISPR-Cas9, have also introduced transformative possibilities, especially for diseases with a clear genetic component. In HD, for instance, CRISPR has been experimentally used to silence the mutant *HTT* gene, which encodes the misfolded huntingtin protein that drives the disease [45]. By selectively editing the gene responsible for this toxic protein, CRISPR offers a path to not only slow disease progression but also potentially correct the underlying genetic defect [46]. In ALS, similar CRISPR applications target mutations in genes like *SOD1* and *C9orf72*, directly addressing the genetic basis of familial cases [47]. The precision of CRISPR technology allows for targeted interventions at the DNA level, effectively altering the trajectory of these diseases in preclinical models [48]. Despite its promise, CRISPR technology faces significant challenges, including off-target effects that may inadvertently alter non-disease-causing genes, and delivery issues that limit its effectiveness in targeting specific brain regions. Ethical considerations, such as the long-term consequences of gene editing and its potential misuse, must also be addressed before widespread clinical application [49].

Epigenetic mechanisms further enrich our understanding of neurodegeneration, revealing how gene expression can be influenced by factors beyond DNA sequence alone. DNA methylation, histone modification, and non-coding RNAs are dynamic regulatory elements that respond to cellular signals and environmental changes [50]. In neurodegenerative diseases, these epigenetic modifications can either exacerbate or protect against neuronal damage. For instance, histone deacetylase (HDAC) inhibitors have been shown to increase the expression of neuroprotective genes and reduce protein aggregation in animal models of HD and ALS. By altering histone acetylation levels, HDAC inhibitors loosen tightly packed chromatin, allowing protective genes to be more actively expressed, which enhances neuron survival [51]. Additionally, non-coding RNAs, particularly microRNAs (miRNAs), play a crucial role in gene regulation by influencing RNA stability and translation. Studies show that specific miRNAs are dysregulated in AD, PD, and ALS, impacting processes such as apoptosis, inflammation, and oxidative stress [52]. These miRNAs are now being explored both as biomarkers for early and precise diagnosis and as therapeutic targets where adjusting miRNA levels could help re-establish cellular equilibrium [53].

Neuroinflammation has also been re-evaluated through a molecular lens, revealing its role as a driving factor rather than merely a byproduct of neurodegeneration [54]. In AD and PD, the brain’s immune cells—microglia—are activated in response to the accumulation of misfolded proteins and damaged cellular components [55]. This activation triggers inflammatory signaling pathways, such as the NF-κB and MAPK pathways, which leads to the release of pro-inflammatory cytokines and chemokines [56]. These molecules, while initially intended to clear cellular debris, sustain a chronic inflammatory state that damages neurons and perpetuates cellular stress [57]. The prolonged activation of microglia thus transforms the brain environment into one that is neurotoxic, hastening neurodegeneration [58]. By targeting specific inflammatory pathways—such as through inhibitors that block cytokine production or through compounds that modulate microglial activation—researchers are exploring anti-inflammatory therapies that could disrupt this cycle and reduce the inflammatory burden in neurodegenerative diseases [59]. Experimental therapies are showing promise in preclinical studies and are moving into clinical trials, marking an exciting step forward in leveraging molecular insights for therapeutic development [60].

Altogether, the molecular understanding of neurodegenerative diseases has set the stage for precision medicine. By integrating data across proteomics, genomics, epigenetics, and immunology, researchers are constructing a detailed molecular map that guides diagnosis and treatment. The vision is not only to identify those at risk through biomarkers but to tailor therapies based on an individual’s unique molecular profile, a personalized approach that holds promise for fundamentally altering the course of neurodegenerative diseases. As molecular techniques continue to evolve, this precision approach is likely to deepen, offering new hope in the fight against these challenging conditions.

### 1.3. Purpose and Scope of the Review

This paper delves into the intricate molecular landscape of neurodegenerative diseases, unraveling the profound connections that bind together conditions like AD, PD, ALS, and HD. Though each of these diseases displays its own distinct clinical features, recent molecular research reveals a fascinating convergence in their underlying mechanisms. Shared hallmarks—such as protein misfolding, mitochondrial dysfunction, oxidative stress, neuroinflammation, and epigenetic changes—suggest that these disorders are bound by common threads, offering a fresh perspective that unites our understanding of neurodegeneration.

At the heart of this exploration is the concept of interconnected cellular dysfunction where one molecular disturbance sets off a cascade of effects across different systems within the cell. For example, a single misfolded protein may trigger oxidative stress, impair mitochondria, and activate inflammatory pathways, creating a feedback loop that further drives protein misfolding. Through this lens, neurodegeneration emerges not as a sequence of isolated failures but as a deeply interconnected network of molecular disruptions, each amplifying the others in a downward spiral toward cellular collapse.

What sets this work apart is its emphasis on the extraordinary translational potential of these molecular insights, opening new doors for diagnostics and therapy. With recent breakthroughs, scientists can now go beyond merely understanding these pathways to actively intervening. Gene-editing techniques like CRISPR-Cas9, for instance, allow for the selective silencing of mutant genes in HD or ALS, while molecular chaperones are showing promise in reshaping misfolded proteins in AD and PD. Once seen as speculative, these approaches are advancing rapidly, with initial studies suggesting that they could redefine the trajectory of these diseases and bring a new level of hope to patients. This paper examines these emerging strategies, shedding light on their potential to reshape the landscape of neurodegenerative treatment.

In addition to therapeutic advances, this analysis highlights how molecular insights are fueling a revolution in personalized diagnostics. The discovery of biomarkers, once a distant ambition, is now making it possible to detect neurodegeneration at its earliest stages. Ultra-sensitive tests now reveal changes in biomarkers like Aβ and tau in AD, or phosphorylated alpha-synuclein in PD, long before symptoms appear, providing a precious opportunity for early intervention. The capacity to diagnose these diseases years before they fully manifest opens the door to proactive, individualized care, marking a shift from late-stage management to pre-emptive action.

In essence, this work bridges the worlds of foundational molecular research and practical application, demonstrating how these insights could drive transformative change. For researchers, it offers a map of interconnected pathways and untapped cross-disease targets. For clinicians, it introduces powerful tools for early diagnosis and precision treatment. For patients and their families, it paints a picture of hope—a future where these insights may lead to early detection, effective intervention, and perhaps, one day, even cures for diseases once considered untreatable. Through this unified approach, the paper captures the extraordinary potential of molecular neurobiology, showing how it is reshaping our understanding and transforming the future of neurodegenerative disease treatment.

## 2. Protein Misfolding and Aggregation

At the core of many neurodegenerative diseases lies a fundamental breakdown in protein folding, a delicate cellular process that normally allows proteins to adopt precise shapes, enabling them to carry out their specific tasks. But in diseases like AD, PD, ALS, and HD, certain proteins misfold and clump together, creating aggregates that disrupt cellular balance and drive the progression of neurodegeneration [61]. These toxic accumulations of Aβ and tau in AD, alpha-synuclein in PD, and huntingtin in HD overwhelm the cell’s natural defenses, activating pathways that, over time, destabilize neurons and lead to their eventual death [62].

Each of these proteins has a unique story—a pathway that explains how they go from functional molecules to toxic agents within the brain [63]. By examining these misfolding pathways in detail, we gain a deeper understanding of the biochemical transformations that lead to disease and, ultimately, how to intervene.

### 2.1. Molecular Pathways of Protein Misfolding

Each neurodegenerative disease is driven by a specific misfolding pathway that turns an otherwise normal protein into a damaging aggregate. These pathways are shaped by a combination of genetic predispositions, cellular stress responses, and environmental factors that, together, push these proteins over the threshold from order to chaos [64]. Below, we explore the molecular journey of Aβ and tau in AD, alpha-synuclein in PD, and huntingtin in HD, focusing on the unique ways they drive neuronal dysfunction.

#### 2.1.1. Amyloid-Beta in Alzheimer’s Disease

In Alzheimer’s disease, the protein Aβ plays a key role in the buildup of toxic plaques. Under normal circumstances, APP, from which Aβ is derived, is cut by enzymes in a way that does not lead to harm [65]. However, in AD, an alternative pathway is activated, causing APP to be cut by beta- and gamma-secretases, which release Aβ fragments, particularly Aβ42. This specific form of Aβ is highly prone to aggregation due to its structure, allowing it to quickly form toxic clusters [66].

Aβ aggregation starts with the formation of small oligomers, or “seed” particles, which have the unique ability to insert themselves into cell membranes. Once embedded, these oligomers disrupt calcium regulation by creating pore-like structures, letting calcium ions flood into the cell [67]. This disruption destabilizes the neuron, affecting processes like synaptic plasticity, which is essential for memory and learning. Moreover, oligomers interact with cell-surface NMDA receptors, overactivating them and leading to excitotoxicity—an excessive calcium influx that damages or kills the cell [68]. Over time, these oligomers join together into larger fibrils and, eventually, amyloid plaques, which set off an immune response that calls in microglia and astrocytes. These immune cells release inflammatory cytokines, creating a neurotoxic environment that further harms nearby neurons [69].

Interestingly, metal ions like zinc and copper bind to Aβ and stabilize its misfolded structure, accelerating the plaque-forming process. Current therapeutic strategies target early steps in this pathway, aiming to block beta- and gamma-secretase activity, chelate metals to slow Aβ aggregation, and use antibodies to clear Aβ oligomers before they accumulate into plaques [70].

#### 2.1.2. Tau Protein Hyperphosphorylation

Tau, another protein central to Alzheimer’s pathology, normally supports the cell’s microtubules, which act like highways within the neuron to transport nutrients and signals. In AD, tau undergoes excessive phosphorylation—a process where phosphate groups attach to the protein [71]. When tau is hyperphosphorylated, it loses its affinity for microtubules, causing them to destabilize. Detached tau proteins then begin to self-associate, creating twisted structures known as paired helical filaments (PHFs) [72]. These filaments grow into neurofibrillary tangles (NFTs) within neurons.

The formation of NFTs disrupts the cell’s internal architecture, leading to failures in nutrient transport and waste removal. Beyond this structural disruption, hyperphosphorylated tau exhibits a prion-like behavior, spreading from cell to cell and seeding misfolding in neighboring neurons. This spread is thought to drive the progression of AD across different brain regions, correlating with worsening cognitive symptoms [73]. Recent research shows that stress responses, like the unfolded protein response (UPR), play a role in tau aggregation. When misfolded proteins accumulate, cells activate the UPR, attempting to restore normal conditions, but this response becomes overactive in AD, perpetuating tau aggregation [74].

Therapeutic approaches for tau pathology include kinase inhibitors, which aim to prevent excessive phosphorylation, and immunotherapies that target tau as it moves between cells. The hope is that by halting tau’s prion-like spread, these therapies can slow or stop the disease’s progression [75].

#### 2.1.3. Alpha-Synuclein in Parkinson’s Disease

In PD, the protein alpha-synuclein misfolds and forms clumps that accumulate inside neurons as Lewy bodies. Alpha-synuclein normally helps regulate dopamine release, but mutations in the *SNCA* gene and environmental toxins can trigger it to misfold. Once misfolded, alpha-synuclein adopts a beta-sheet structure that is prone to aggregation, forming oligomers that are highly toxic [76].

These oligomers of alpha-synuclein integrate into cellular membranes, disrupting their integrity by creating pores. These pores allow unregulated calcium influx, which disrupts cellular homeostasis and strains mitochondria, leading to excessive production of ROS and activation of apoptosis, or programmed cell death [77]. Dopaminergic neurons are especially susceptible to this toxicity because they rely heavily on energy-demanding processes and have relatively low antioxidant defenses. As oligomers grow into fibrils and Lewy bodies, they interfere with cellular recycling processes like autophagy, causing damaged proteins and organelles to accumulate within the cell [78].

Furthermore, alpha-synuclein exhibits prion-like properties, allowing misfolded forms to travel between neurons. Extracellular alpha-synuclein is taken up by neighboring cells, inducing aggregation and spreading pathology across brain regions in a way that mirrors PD’s clinical progression [79]. Therapies for PD focus on stabilizing alpha-synuclein in its native form, using antibodies to clear misfolded proteins, and enhancing autophagy to improve the cell’s ability to degrade alpha-synuclein aggregates [42].

#### 2.1.4. Mutant Huntingtin in Huntington’s Disease

Huntington’s disease arises from a mutation in the *HTT* gene, leading to an abnormal expansion of CAG repeats that encode a polyglutamine (polyQ) tract in the huntingtin protein. This extended polyQ region makes the mHTT sticky and prone to forming toxic aggregates [23]. The aggregation of mHTT is particularly harmful in the striatum, a brain region critical for motor function, where it interferes with essential cellular processes [80].

Mutant huntingtin disrupts transcription by binding to transcription factors and coactivators, interfering with the normal expression of genes essential for neuronal survival [81]. mHTT also impairs autophagy, the cell’s main pathway for clearing damaged components. By disrupting autophagy, mHTT prevents the cell from effectively removing itself, allowing damaged proteins and aggregates to accumulate. Additionally, mHTT interacts with mitochondria, impairing their function and leading to energy deficits, which compounds cellular stress [82].

Therapeutic strategies for HD include approaches that silence mutant *HTT* expression through RNA interference and antisense oligonucleotides (ASOs), reducing the production of the toxic protein. Other therapies aim to enhance autophagy, helping cells clear mHTT aggregates or stabilize mitochondrial function, thereby alleviating some of the cellular damage caused by mHTT [83].

Each of these pathways reveals a unique aspect of protein misfolding, but together, they paint a picture of how small molecular changes can lead to devastating consequences. Misfolded proteins disrupt essential cellular processes, overwhelm protective pathways, and, as they accumulate, create an environment where neurons struggle to survive [84]. By targeting these early stages of misfolding and aggregation, researchers hope to develop therapies that can interrupt the cascade of cellular dysfunction, offering new avenues for treating neurodegenerative diseases at their source [85].

The following table (Table 1) provides an overview of the fundamental molecular mechanisms associated with neurodegenerative diseases, emphasizing high-impact studies that have shaped our understanding of each mechanism’s role. The table categorizes key processes such as protein misfolding, neuroinflammation, and mitochondrial dysfunction, linking each mechanism to specific therapeutic targets. By synthesizing data from highly cited research, this table illustrates the multifaceted nature of neurodegenerative disease pathology and highlights current therapeutic approaches aimed at modulating these critical molecular pathways.

### 2.2. Cellular Consequences of Protein Aggregation

In neurodegenerative diseases, the buildup of misfolded proteins within neurons does much more than just accumulate; it disrupts the cell’s natural rhythm and resilience. These protein clumps overwhelm the cell’s protective systems, overloading its capacity to handle protein folding, degradation, and calcium balance [96]. Neurons, with their intricate structure and high energy needs, are especially vulnerable to this kind of stress. Over time, the presence of these aggregates creates a toxic environment, setting off a cascade of dysfunction that leaves neurons struggling to survive. Understanding how protein aggregation disturbs cellular balance and proteostasis reveals the deep-seated challenges neurons face, highlighting how these disruptions ultimately push cells toward degeneration.

#### 2.2.1. Disruption of Cellular Homeostasis

One of the first and most severe impacts of protein aggregation in neurodegenerative diseases is the stress it places on the endoplasmic reticulum (ER), the cell’s central station for protein folding, lipid synthesis, and calcium storage. The ER is finely tuned to detect misfolded proteins, and when these start piling up, it activates the UPR, a defense mechanism designed to temporarily boost the cell’s folding capacity and promote the clearance of misfolded proteins [97]. But when protein misfolding is constant, as it is in these diseases, the UPR remains switched on, eventually shifting from a protective to a harmful role as the cell’s resources are stretched beyond capacity [98].

In AD, Aβ oligomers wreak havoc on calcium regulation within the ER. These toxic clumps latch onto calcium channels in the ER membrane, causing calcium to spill into the cytoplasm and disrupt essential signaling pathways that neurons rely on for communication [99]. As calcium floods into the cell, it also overloads mitochondria, which attempt to buffer the imbalance but become overwhelmed, swelling and releasing ROS. These excess ROS, in turn, damage cellular components and push the neuron closer to apoptosis, or programmed cell death.

PD presents a similar situation, with alpha-synuclein aggregates triggering stress responses within the ER. These misfolded proteins bind to BiP, an ER chaperone that plays a central role in managing protein folding [100]. With BiP compromised, the ER is left unable to handle the growing load of misfolded proteins, leading to prolonged UPR activation. This chronic stress response triggers inflammatory cytokines, which invite immune cells and create an environment of neuroinflammation [101]. Over time, ER stress induces CHOP, a transcription factor that activates pro-apoptotic genes, setting the neuron on a path toward programmed death [102].

These disruptions in calcium regulation and persistent ER stress create a cycle that is particularly harmful in neurons. As calcium leaks from the ER, it not only damages mitochondria but also erodes the neuron’s ability to sustain essential functions. This constant strain weakens the cell’s defenses, leaving neurons less able to handle additional stress and making degeneration almost inevitable [103].

#### 2.2.2. Impairment of Proteostasis

Proteostasis, or the balance of protein maintenance, is a delicate process essential for cellular health, especially in neurons where proper protein turnover is crucial for survival [104]. This balance relies on two main pathways: the ubiquitin–proteasome system (UPS) and the autophagy–lysosomal pathway, both of which work together to identify, break down, and clear misfolded proteins and cellular waste [63]. In neurodegenerative diseases, however, these systems are overwhelmed, leaving the cell unable to handle the mounting accumulation of damaged proteins, which results in a toxic buildup that further stresses neurons [105].

Ubiquitin–Proteasome System Dysfunction

The UPS is responsible for identifying and degrading small, damaged, or short-lived proteins. Proteins marked for disposal are tagged with ubiquitin molecules, directing them to the proteasome where they are broken down for recycling. But in neurodegenerative diseases, protein aggregates inhibit the UPS, leaving ubiquitinated proteins to accumulate within the cell [106].

In Alzheimer’s disease, tau aggregates directly interfere with the proteasome’s function. When hyperphosphorylated, tau binds to the proteasome’s 20S core, blocking the entry of proteins marked for degradation. As these proteins back up, they form secondary aggregates, adding further strain to the cell’s quality control systems. This UPS impairment disrupts normal cellular functions that rely on regulated protein turnover, leading to additional stress on the neuron [107].

In Parkinson’s disease, alpha-synuclein clogs the proteasome by obstructing its entry points, preventing ubiquitinated proteins from being processed. Studies show that alpha-synuclein oligomers interact with the 19S regulatory particle, blocking substrate entry and leading to a buildup of damaged proteins within dopaminergic neurons [108]. This failure of the UPS not only accelerates the aggregation of alpha-synuclein but also adds to the cellular backlog, creating a cycle of rising toxicity and stress within the cell [109].

Autophagy–Lysosomal Pathway Defects

The autophagy–lysosomal pathway handles the degradation of larger protein aggregates and damaged organelles. Through autophagy, cells create vesicles known as autophagosomes that capture cellular debris, which are then fused with lysosomes to break down and recycle the contents [110]. In neurodegenerative diseases, however, this pathway becomes impaired, preventing the cell from removing toxic aggregates [111].

In HD, the mHTT protein directly impairs autophagy by sequestering key receptors like p62 and LC3, which are essential for identifying and capturing cargo for degradation. When these receptors are trapped within mHTT aggregates, autophagosomes cannot form properly, leaving toxic aggregates and damaged organelles to build up within the cell [112]. This accumulation strains the mitochondrial function, raises oxidative stress, and accelerates cellular decline, contributing to the characteristic cell death seen in HD [113,114].

In PD, alpha-synuclein disrupts lysosomal acidification, which is crucial for activating lysosomal enzymes [115]. Without the proper acidic environment, these enzymes cannot degrade alpha-synuclein fibrils or other cellular waste effectively. This impairment leads to a backlog of undegraded material in autophagosomes, which then leak into the cell, increasing oxidative stress and triggering inflammation [116]. Alpha-synuclein also affects chaperone-mediated autophagy (CMA), a pathway that degrades proteins through direct lysosomal delivery [117]. By blocking LAMP-2A, a critical lysosomal receptor, alpha-synuclein further reduces the cell’s capacity to clear misfolded proteins, leading to a progressive toxic buildup that further destabilizes the neuron [118].

In ALS, TDP-43 aggregates block the fusion of autophagosomes with lysosomes, preventing the clearance of cellular waste [119]. TDP-43 also disrupts mitophagy, a form of autophagy specifically responsible for clearing damaged mitochondria [104]. The inability to clear dysfunctional mitochondria leads to a buildup of ROS and depletes ATP reserves, creating an energy crisis that strains motor neurons, particularly susceptible to high energy demands, and accelerates their degeneration [120].

Altogether, the failure of proteostasis—due to the impairment of both UPS and autophagy–lysosomal pathways—leaves neurons in a state of toxic overload, unable to clear out the cellular debris that continues to accumulate [121]. This chronic buildup of damaged proteins and organelles disrupts metabolic processes, heightens oxidative stress, and creates a toxic cellular environment that gradually drives neurons to the brink [122].

The breakdown in cellular homeostasis and proteostasis due to protein aggregation creates a cycle of stress that neurons cannot easily escape. As these cells struggle with constant misfolded proteins, oxidative damage, and metabolic instability, they become increasingly vulnerable to degeneration [123]. Understanding these complex interactions opens up new avenues for therapeutic research, as scientists work to reinforce these protective systems and develop treatments that might slow or even stop neurodegenerative progression at its roots.

### 2.3. Molecular Therapeutic Strategies

With the advances in understanding the molecular foundations of neurodegenerative diseases, there’s now a push toward treatments that intervene directly at the roots of cellular dysfunction [124]. Traditional treatments often only address symptoms, but these novel approaches target the misfolded proteins and disrupted cellular pathways at the heart of diseases like Alzheimer’s, Parkinson’s, ALS, and Huntington’s [125]. These therapies focus on stopping protein aggregation, enhancing cellular protein clearance, and supporting proper protein folding with molecular chaperones. Each strategy aims to restore cellular health, reduce toxic accumulations, and protect neurons from degeneration [126].

#### 2.3.1. Inhibition of Protein Aggregation

A primary approach to neurodegenerative therapy involves small molecules designed to stop proteins from aggregating in the first place. These molecules work by binding to the misfolded proteins, either stabilizing them in a less toxic form or preventing them from clustering into larger, harmful assemblies.

Targeting Amyloid-Beta and Tau in Alzheimer’s Disease

In AD, the aggregation of Aβ and tau is a well-known feature of disease pathology. Researchers have developed small molecules that bind to Aβ monomers, preventing them from joining into toxic oligomers that interfere with synaptic function [127,128]. One such compound, tramiprosate, binds to Aβ and stabilizes it in a way that blocks its self-assembly, while EPPS (4-(2-hydroxyethyl)-1-piperazinepropanesulfonic acid) has been shown to actually break down existing amyloid plaques in preclinical studies, suggesting potential to reduce Aβ burden in the brain [129].

Similarly, tau aggregation inhibitors focus on preventing tau from detaching from microtubules and forming neurofibrillary tangles. Compounds like LMTM (leuco-methylthioninium bis(hydromethanesulfonate)) stabilize tau’s normal structure, thereby reducing its tendency to aggregate into the paired helical filaments that comprise neurofibrillary tangles [130]. LMTM works by inhibiting tau phosphorylation, which normally triggers its detachment from microtubules, and has shown promise in preclinical trials for preserving synaptic health and slowing cognitive decline [131,132].

The following figure (Figure 1) provides a visual overview of the steps involved in Aβ formation and plaque accumulation. It illustrates how APP is cleaved by β- and γ-secretases to release Aβ peptides, which, subsequently, aggregate in the extracellular space, forming the amyloid plaques that are characteristic of Alzheimer’s disease pathology.

Preventing Alpha-Synuclein Aggregation in Parkinson’s Disease

In PD, alpha-synuclein misfolding leads to the formation of Lewy bodies, which disrupt cellular communication and dopamine production. Small molecules like anle138b interact with alpha-synuclein early in the misfolding process, preventing it from forming toxic fibrils [133]. By stabilizing alpha-synuclein, these molecules reduce the buildup of Lewy bodies, preserving cellular function [134]. Additionally, NPT200-11 has shown the potential to reduce alpha-synuclein pathology by binding to alpha-synuclein oligomers, keeping them from forming the toxic aggregates that damage dopaminergic neurons [135].

Inhibiting Mutant Huntingtin Aggregation in Huntington’s Disease

HD is marked by the aggregation of protein mHTT with extended polyQ sequences, leading to toxic inclusions in neurons [136]. Compounds such as EGCG (epigallocatechin gallate) and C2-8 target these polyQ regions, reducing mHTT’s tendency to self-aggregate and potentially enhancing clearance of these protein clusters. Studies show that EGCG binds directly to huntingtin, stabilizing its structure and preventing aggregation [137]. This stabilization reduces cellular stress, making these inhibitors promising candidates for therapies targeting the root pathology of HD [138].

#### 2.3.2. Enhancing Protein Clearance Pathways

The UPS and autophagy–lysosomal pathway are the cell’s main methods for clearing out damaged and misfolded proteins. In neurodegenerative diseases, these systems become overwhelmed by toxic accumulations [139]. Therapies aimed at boosting these pathways work to enhance the cell’s ability to clear misfolded proteins before they cause harm [140,141].

Enhancing the Ubiquitin–Proteasome System

The UPS handles the breakdown of short-lived or damaged proteins. In this system, proteins tagged with ubiquitin are directed to the proteasome for degradation. However, when misfolded proteins accumulate beyond the system’s capacity, the UPS becomes overloaded [142]. Proteasome activators, like 3,4-dimethoxychalcone, show promise in restoring UPS activity in AD models where they promote the degradation of tau and prevent its buildup within neurons. By enhancing proteasome activity, these compounds can alleviate some of the toxic load within cells, reducing cellular stress and slowing disease progression.

Selective enhancement of E3 ligase activity is another approach to improving UPS efficiency. E3 ligases add ubiquitin tags to specific misfolded proteins, marking them for degradation. In ALS and AD, where proteins like TDP-43 and tau form damaging aggregates, activating specific E3 ligases helps the UPS selectively target these toxic proteins, enabling the cell to manage misfolded proteins more effectively and prevent them from accumulating in the first place [143].

Stimulating Autophagy and the Lysosomal Pathway

Autophagy is essential for clearing larger aggregates and damaged cellular components. By inducing autophagy, cells can encapsulate aggregates within autophagosomes, which then fuse with lysosomes to degrade their contents. Rapamycin, a well-known autophagy inducer, promotes the formation of autophagosomes by inhibiting mTOR, a regulator of cellular growth that normally suppresses autophagy [144]. In HD models, rapamycin has shown promise in clearing mutant huntingtin aggregates, thus improving cellular health and reducing motor symptoms [145].

In Parkinson’s disease, lysosomal dysfunction plays a significant role in the buildup of alpha-synuclein. Ambroxol, a compound that promotes lysosomal acidification, restores the activity of lysosomal enzymes like glucocerebrosidase, which is essential for breaking down alpha-synuclein fibrils [146]. By improving lysosomal function, ambroxol enhances the cell’s ability to manage protein overload, which is essential for reducing cellular toxicity and preserving dopaminergic neuron function [147].

#### 2.3.3. Molecular Chaperone Therapies

Molecular chaperones, particularly heat shock proteins (HSPs), are essential for helping proteins fold correctly and preventing misfolding [148]. In neurodegenerative diseases, chaperone support has become an area of interest as enhancing chaperone levels can stabilize misfolded proteins and prevent aggregation.

Supporting Chaperones in Alzheimer’s and Parkinson’s Disease

In Alzheimer’s disease, the chaperones HSP70 and HSP90 are known to interact with tau, assisting in proper folding and preventing tangle formation. Arimoclomol, a drug that stimulates heat shock protein production, increases levels of HSP70, which stabilizes tau and reduces its aggregation potential. Studies suggest that arimoclomol’s effects on HSP70 also help guide misfolded tau toward degradation pathways, reducing the buildup of neurofibrillary tangles [149].

In Parkinson’s disease, small molecules like YM-1 help upregulate HSP70 and HSP27, which play key roles in stabilizing alpha-synuclein and preventing it from forming toxic oligomers. By promoting the expression of these chaperones, YM-1 reduces alpha-synuclein toxicity and helps maintain cellular integrity. This approach has shown promise in models of PD, suggesting that chaperone-based therapies could slow the spread of alpha-synuclein pathology within the brain [150].

Chaperone Enhancement in ALS and Huntington’s Disease

In ALS, where the misfolding of TDP-43 and SOD1 is common, enhancing chaperone activity helps manage these proteins. Arimoclomol has demonstrated potential in increasing HSP levels, aiding the refolding or degradation of SOD1 aggregates and supporting neuronal health. The effect of arimoclomol in ALS models includes reduced protein aggregation, improved cell survival, and delayed disease progression [151].

In HD, chaperones like HSP40 and HSP70 have shown potential for reducing huntingtin aggregation. Geldanamycin, an HSP90 inhibitor, promotes HSP70-mediated pathways that guide mutant huntingtin toward degradation. This shift helps reduce the toxic burden of mHTT, with studies showing improved motor function and cellular resilience in HD models [152]. By supporting chaperones that keep proteins in proper shape or direct them toward clearance, these therapies offer a valuable strategy for countering the molecular disruptions of neurodegenerative diseases [153].

The therapeutic landscape for neurodegenerative diseases is rapidly evolving, with approaches that go beyond symptom relief to address the molecular drivers of disease progression. By preventing protein aggregation, enhancing cellular clearance systems, and leveraging chaperones for protein stability, these strategies aim to protect neurons from the cascading effects of misfolded proteins [154]. Though many of these treatments remain experimental, their potential to directly address the cellular imbalances that fuel neurodegeneration holds promise for interventions that could slow or halt the progression of diseases like Alzheimer’s, Parkinson’s, ALS, and Huntington’s, offering a transformative impact on patient outcomes.

## 3. Neuroinflammation and Molecular Immune Responses

Neuroinflammation, while a natural defense mechanism, takes on a harmful role in neurodegenerative diseases. In these conditions, inflammation becomes a chronic and self-perpetuating process, driven largely by immune cells in the brain, such as microglia and astrocytes. Initially activated to clear damaged neurons and misfolded proteins, these cells eventually maintain a state of prolonged activation [155]. This chronic inflammation leads to a continuous release of pro-inflammatory molecules, cytokines, and reactive oxygen species, creating a toxic environment that accelerates neuronal damage rather than alleviating it [156].

At the molecular level, neuroinflammation is fueled by specialized immune receptors, known as pattern recognition receptors (PRRs), which detect signs of cellular distress. These receptors respond to molecular signals called damage-associated molecular patterns (DAMPs), which are released by stressed or dying cells [157]. Together, PRRs and DAMPs drive the inflammatory response, transforming short-term immune activation into a long-term source of damage in neurodegenerative diseases. Understanding the molecular dynamics of these interactions offers a promising direction for therapeutic strategies that could disrupt the cycle of inflammation and protect neurons from further harm [158].

### 3.1. Molecular Triggers of Neuroinflammation

In neurodegenerative diseases, immune receptors on brain cells sense abnormalities, including protein aggregates and cellular debris. These abnormalities are detected by PRRs, which serve as cellular “sensors” that recognize DAMPs—molecules released by damaged or dying cells as distress signals. When PRRs are activated by DAMPs, they initiate inflammatory pathways that contribute to prolonged neuroinflammation [159]. The main PRRs implicated in this process include Toll-like receptors (TLRs), NOD-like receptors (NLRs), and the receptor for advanced glycation end-products (RAGE). These receptors recognize and respond to specific DAMPs, such as HMGB1, ATP, and mtDNA, amplifying immune responses that, over time, contribute to neuronal loss [160].

Pattern Recognition Receptors (PRRs)

PRRs are critical to the brain’s immune response as they specialize in detecting signs of cell damage and stress. These receptors, including TLRs, NLRs, and RAGE, are integral to the inflammatory processes seen in neurodegenerative diseases. Each type of PRR detects unique molecular patterns and activates distinct inflammatory pathways [161].

TLRs are cell-surface receptors that recognize extracellular danger signals. For instance, TLR4, a key player in AD, binds to Aβ aggregates and activates the NF-κB signaling pathway, which leads to the production of pro-inflammatory cytokines like interleukin-1β (IL-1β) and tumor necrosis factor-alpha (TNF-α) [162]. In PD, TLR2 is activated by alpha-synuclein aggregates, setting off a cascade involving MAPK and NF-κB pathways, which increase the release of inflammatory mediators. Research suggests that blocking TLR2 and TLR4 activity could mitigate inflammatory damage, indicating their potential as therapeutic targets [163].

NLRs and the NLRP3 inflammasome respond to intracellular signals of cellular distress. The NLRP3 inflammasome, a multi-protein complex, is activated by factors like mitochondrial dysfunction, ROS, and protein aggregates such as Aβ and alpha-synuclein [164]. Upon activation, NLRP3 facilitates the activation of caspase-1, which processes cytokines like IL-1β and IL-18 into their active forms, intensifying the inflammatory response. In AD, NLRP3 activation near amyloid plaques contributes to neuron damage, while in PD, cytosolic alpha-synuclein activates NLRP3, promoting sustained inflammation [165]. Inhibiting NLRP3 has shown promising results in preclinical studies as it reduces neuroinflammation and slows disease progression, highlighting its potential as a therapeutic target [166].

RAGE recognizes advanced glycation end-products (AGEs) as well as other DAMPs, including Aβ and HMGB1. Once activated, RAGE promotes inflammation through NF-κB signaling, increasing cytokine production and ROS release. In AD, RAGE binding with Aβ enhances neurotoxic pathways and amyloid plaque formation, contributing to cognitive decline [167]. In ALS, RAGE activation by AGEs and oxidative stress compounds motor neuron degeneration. Experimental RAGE inhibitors are being studied for their potential to disrupt these harmful interactions and reduce inflammation, representing another promising direction for therapeutic intervention [168].

Damage-Associated Molecular Patterns (DAMPs)

DAMPs are endogenous molecules that act as distress signals, alerting the immune system to cellular damage. Released from injured cells, DAMPs activate PRRs on microglia and astrocytes, maintaining a pro-inflammatory state that becomes chronic in neurodegenerative diseases. Key DAMPs in these diseases include HMGB1, extracellular ATP, and mtDNA, each contributing uniquely to the inflammation seen in neurodegeneration [169,170].

High mobility group box 1 (HMGB1) is a nuclear protein that helps regulate chromatin structure, but when released outside cells, it acts as a DAMP. In neurodegenerative diseases, HMGB1 is elevated in affected brain regions, such as the hippocampus in AD and motor neurons in ALS [171]. Once outside the cell, HMGB1 binds to receptors like TLR4 and RAGE, activating signaling pathways that lead to cytokine release and oxidative stress [172]. This sustained cytokine release disrupts synaptic function and accelerates neuronal death, particularly in areas surrounding amyloid plaques in AD. Blocking HMGB1 signaling has shown potential in reducing neuroinflammation and protecting against neurodegeneration [173].

Extracellular ATP and purinergic receptors serve as potent inflammatory signals when released from cells during injury. ATP binds to purinergic receptors like P2X7 on microglia, activating inflammasomes such as NLRP3, which further amplify inflammatory responses by releasing IL-1β and other cytokines [174]. In PD, elevated extracellular ATP levels enhance alpha-synuclein toxicity by activating P2X7. Experimental approaches that block P2X7 signaling show promise in reducing ATP-driven neuroinflammation and protecting dopaminergic neurons from sustained damage [175].

mtDNA is typically contained within mitochondria, but when mitochondria are damaged, mtDNA leaks into the extracellular space where it acts as a powerful DAMP [176]. Once outside the mitochondria, mtDNA binds to receptors like TLR9, which recognizes specific unmethylated DNA sequences. In ALS, mtDNA released from injured motor neurons triggers TLR9 activation in nearby glial cells, contributing to chronic inflammation and oxidative stress. Research into blocking TLR9 activation or limiting mtDNA release suggests potential therapeutic strategies to reduce inflammation in diseases driven by mitochondrial dysfunction [169].

Together, PRRs and DAMPs create a self-sustaining cycle of neuroinflammation in neurodegenerative diseases where prolonged immune activation results in progressive neuronal loss [177]. By targeting these molecular mechanisms, researchers are working to develop therapies that can break the cycle of chronic inflammation, protect neurons, and slow the progression of neurodegenerative diseases, offering hope for interventions that directly address the underlying drivers of neuroinflammatory damage [178].

### 3.2. Microglial Activation States

Microglia, the brain’s resident immune cells, are essential for maintaining neural health and responding to injury. In neurodegenerative diseases, these cells often shift from protective roles to states that exacerbate inflammation and contribute to neuronal damage [179]. Microglial activation is dynamic, encompassing a spectrum of responses, but is often simplified into two main functional states: pro-inflammatory “M1” and anti-inflammatory “M2”. This shift from protective to harmful roles is influenced by signals within the disease environment, making microglial activation a central focus for potential therapeutic intervention in neurodegenerative diseases [180].

The following figure (Figure 2) illustrates the differentiation of monocytes into M0 macrophages, which can then polarize into M1 or M2 phenotypes depending on environmental cues. By understanding the molecular pathways that drive macrophage polarization, researchers are exploring strategies to promote M2-like states in neurodegenerative diseases, aiming to harness their anti-inflammatory and protective functions to mitigate disease progression and support neuronal health.

#### 3.2.1. M1 vs. M2 Phenotypes

The M1 microglial phenotype is characterized by pro-inflammatory activity. When microglia adopt this state, they release high levels of cytokines such as IL-1β, TNF-α, and interleukin-6 (IL-6), as well as ROS and nitric oxide (NO) [181]. While these responses help protect the brain from pathogens and clear cellular debris, in neurodegenerative conditions, sustained M1 activation creates a neurotoxic environment [182]. Misfolded proteins, cellular stress, and inflammatory signals common in AD, PD, and ALS continuously fuel this state, overwhelming the brain’s capacity to maintain homeostasis and contributing to ongoing neuronal loss.

Monocyte differentiation into macrophages and subsequent polarization to M1 or M2 phenotypes is driven by specific molecular signals. M1 polarization is induced by pro-inflammatory signals like interferon-gamma (IFN-γ) and lipopolysaccharides (LPS), leading to a pro-inflammatory state. Conversely, M2 polarization is promoted by anti-inflammatory cytokines such as IL-4 and IL-13, fostering tissue repair and resolution of inflammation [183].

On the other end of the spectrum, the M2 phenotype is associated with anti-inflammatory functions and tissue repair [184]. M2 microglia release cytokines such as interleukin-10 (IL-10) and transforming growth factor-beta (TGF-β), which help resolve inflammation and support neuroprotection [185]. They also promote phagocytosis, aiding in the clearance of misfolded proteins and cellular debris. In neurodegenerative diseases, however, the shift toward an M2 phenotype is often suppressed, limiting the brain’s ability to repair and recover [186]. Therapeutic strategies that promote M2 activation are being investigated as a way to reduce inflammation and protect against progressive neuronal damage [187]. PPAR-γ agonists and IL-10 enhancers are being explored to shift microglia toward an M2 phenotype, while agents targeting phagocytic pathways, such as TREM2 activators, aim to enhance the clearance of neurotoxic debris. By rebalancing microglial states, these approaches seek to alleviate chronic inflammation and create a supportive environment for neuronal repair [188].

#### 3.2.2. Signaling Pathways in Activation

Microglial activation depends on intricate signaling networks that drive microglia toward either pro-inflammatory or protective states [189]. Among these, the NF-κB and MAPK pathways are key players, shaping microglial responses in the presence of pathological signals [190].

The NF-κB (nuclear factor kappa-light-chain-enhancer of activated B cells) pathway is a primary regulator of inflammation in microglia. Activated by stimuli such as Aβ in AD and alpha-synuclein in PD, NF-κB translocates to the nucleus where it initiates the production of pro-inflammatory molecules like TNF-α, IL-1β, and IL-6 [191]. While NF-κB activity helps contain early threats, chronic activation in disease conditions drives a long-term inflammatory response. Modulating this pathway to reduce prolonged NF-κB signaling is being explored as a strategy to limit neuroinflammation [192].

The MAPK (mitogen-activated protein kinase) pathway is another critical pathway involved in microglial activation. Activated by external stress signals, the MAPK pathway includes kinases such as ERK, JNK, and p38, which regulate microglial responses to DAMPs [193]. For example, the activation of p38 MAPK has been linked to increased production of IL-1β and TNF-α, promoting inflammation in neurodegenerative disease contexts [56]. Targeting specific components of the MAPK pathway, especially p38, offers a potential therapeutic avenue to reduce pro-inflammatory microglial responses while preserving protective functions [194].

Other signaling molecules also influence the M1/M2 balance. Peroxisome proliferator-activated receptors (PPARs), particularly PPAR-gamma, are known to encourage M2 polarization by inhibiting NF-κB and suppressing the release of inflammatory cytokines [195]. Similarly, STAT6 (signal transducer and activator of transcription 6) activation supports M2 differentiation, enhancing the expression of anti-inflammatory molecules like IL-10 [196]. Therapies that target PPAR-gamma or STAT6 to promote M2 activation hold promise for shifting microglial responses away from sustained inflammation and toward protective roles in the brain [197].

By better understanding the molecular signals that drive microglial states, researchers aim to restore the balance between M1 and M2 functions, reducing the harmful inflammation that accelerates neurodegeneration. Encouraging a shift from the pro-inflammatory M1 state to the neuroprotective M2 state is a promising strategy, with the potential to alleviate chronic inflammation and enhance neural repair.

### 3.3. Cytokine Networks

Cytokines, small but powerful signaling proteins released by immune cells, orchestrate inflammation and communication within the brain’s immune landscape. In healthy conditions, cytokine release is balanced, with pro-inflammatory signals initiating defense responses and anti-inflammatory signals resolving them [198]. However, in neurodegenerative diseases like AD, PD, ALS, and HD, this balance is disrupted. Chronic inflammation arises as pro-inflammatory cytokines dominate, fueling cellular stress and neuronal damage [199]. These intricate cytokine networks—encompassing both inflammatory and protective signals—reveal insights into the persistence of inflammation in these conditions and offer potential targets for modulating immune responses to protect neurons.

#### 3.3.1. Pro-Inflammatory Cytokines

In neurodegenerative diseases, the persistent release of pro-inflammatory cytokines becomes a source of cellular distress. While intended to respond to injury or infection, these cytokines, when chronically elevated, create an environment that accelerates neurodegeneration. Among the most influential pro-inflammatory cytokines in these diseases are TNF-α, IL-1β, and IL-6 [200].

TNF-α is a central player in neuroinflammation, commonly elevated in affected brain regions. TNF-α binds to receptors on neurons and glial cells, activating signaling pathways like NF-κB, which further amplify inflammatory responses. In AD, TNF-α has been shown to impair synaptic function, while in PD, it contributes to the degeneration of dopaminergic neurons [55]. Animal studies suggest that blocking TNF-α signaling can alleviate inflammation and reduce neurotoxicity, and therapies targeting this cytokine are being developed as a means to mitigate its detrimental effects [201].

IL-1β is another potent pro-inflammatory mediator produced by activated microglia and astrocytes. It increases the permeability of the blood–brain barrier (BBB), allowing peripheral immune cells to enter the brain, thereby compounding inflammation [202]. In AD, IL-1β is released in response to Aβ plaques, intensifying neuronal damage around these aggregates. In ALS, IL-1β contributes to BBB breakdown and amplifies immune activity in ways that are particularly harmful to motor neurons. Targeting IL-1β or its pathways could offer a promising route to control neuroinflammation and protect the BBB in these diseases [203].

IL-6 has both pro-inflammatory and anti-inflammatory roles, but in neurodegenerative contexts, it typically promotes inflammation. Elevated IL-6 levels in cerebrospinal fluid and brain tissues of AD and PD patients correlate with increased glial activation and the release of other inflammatory signals [204]. By activating the JAK/STAT3 pathway, IL-6 sustains a feedback loop of inflammation, particularly detrimental in ALS where it activates astrocytes and microglia, accelerating motor neuron degeneration. Researchers are investigating therapies that inhibit IL-6 signaling as a potential means to reduce neuroinflammation and preserve neuronal health [205,206].

#### 3.3.2. Anti-Inflammatory Mediators

In contrast to the destructive role of pro-inflammatory cytokines, anti-inflammatory mediators act to resolve inflammation and promote cellular repair. These cytokines work to counterbalance inflammation, yet in neurodegenerative diseases, their levels are often inadequate to mitigate chronic inflammatory responses. Two key anti-inflammatory cytokines with neuroprotective potential are IL-10 and TGF-β [207,208].

IL-10, produced by microglia, astrocytes, and infiltrating immune cells, inhibits the release of pro-inflammatory cytokines and reduces microglial activation via the STAT3 pathway, which represses inflammatory gene transcription [209]. IL-10’s neuroprotective role is evident in its ability to limit excessive immune responses, yet its levels are often insufficient to counteract sustained inflammation in neurodegenerative diseases [210]. Strategies to enhance IL-10 activity are being explored as a way to promote an anti-inflammatory state in microglia, potentially slowing disease progression by creating a more protective environment for neurons [211].

TGF-β serves diverse functions within the CNS, modulating immune activity, promoting tissue repair, and helping maintain BBB integrity. By limiting pro-inflammatory cytokine production and supporting the M2 (anti-inflammatory) microglial phenotype, TGF-β plays a key role in controlling inflammation [212]. However, TGF-β signaling is often disrupted in conditions like AD and PD, diminishing its regulatory effects on immune activity. Efforts to restore or enhance TGF-β signaling are underway, with the aim of reinforcing the brain’s natural anti-inflammatory mechanisms and providing support for neuronal survival [213].

The interplay between pro-inflammatory and anti-inflammatory cytokines shapes the immune environment within the neurodegenerating brain. When pro-inflammatory signals predominate, inflammation becomes a chronic, self-sustaining force, damaging neurons and accelerating disease progression [214]. Understanding these cytokine networks opens pathways for therapeutic interventions that rebalance inflammatory responses, reduce neurotoxicity, and promote protective immune activity. Targeting specific cytokines or modulating signaling pathways offers hope for treatments that could mitigate inflammation, preserve neuron function, and slow the advancement of neurodegenerative diseases.

### 3.4. Molecular Targets for Anti-Inflammatory Therapy

Chronic inflammation within the brain drives neurodegeneration in diseases. Anti-inflammatory therapies that target specific molecular pathways aim to restore immune balance, reduce neurotoxicity, and protect neurons [215]. By inhibiting harmful cytokine production, adjusting microglial activation, and enhancing protective signaling, these therapies offer a focused approach to addressing inflammation as a root cause of neurodegeneration [216].

#### 3.4.1. Inhibitors of Cytokine Production

Targeting the overproduction of pro-inflammatory cytokines is central to managing neuroinflammation. Inhibiting key pathways such as NF-κB and JAK/STAT offers a precise means to curtail cytokine signaling while preserving essential immune functions [217].

NF-κB Pathway Inhibitors: NF-κB is a key regulator of inflammation, often overactivated in neurodegenerative conditions. Small molecule inhibitors targeting IκB kinase (IKK), an activator of NF-κB, aim to reduce cytokine release (e.g., TNF-α, IL-1β) linked to neuronal damage in AD and PD [218]. These inhibitors selectively suppress NF-κB without broadly dampening immune response, showing potential in animal models for decreasing inflammation and preserving neuron function [219].

JAK/STAT Pathway Modulation: The JAK/STAT pathway, particularly STAT3 activation by IL-6, sustains inflammation in neurodegenerative disease. Janus kinase (JAK) inhibitors such as tofacitinib, which have been effective in autoimmune disorders, are being studied in neurodegenerative contexts [220]. These inhibitors target the JAK/STAT axis to selectively mitigate cytokine signaling and its downstream inflammatory effects, offering a way to protect neurons from chronic inflammation [221].

#### 3.4.2. Modulation of Microglial Activation

Microglia, the brain’s immune cells, play a central role in neuroinflammation. Therapies that shift microglia from a pro-inflammatory state to a neuroprotective profile hold promise for alleviating inflammation and supporting tissue repair [222].

PPAR-gamma Agonists: PPAR-γ agonists, such as pioglitazone, encourage microglia to adopt a protective role by activating PPAR-γ, a nuclear receptor regulating inflammation [223]. These agonists reduce the release of inflammatory cytokines and improve microglial clearance of toxic aggregates. In AD and PD models, PPAR-γ activation reduces oxidative stress and enhances microglial function, suggesting its potential for restoring immune balance in the brain [224].

Colony-Stimulating Factor 1 Receptor (CSF1R) Inhibitors: CSF1R is critical for microglial survival and activation [225]. CSF1R inhibitors, such as PLX3397, selectively modulate microglial activity, reducing their neurotoxic effects without eliminating them. These inhibitors limit excessive microglial activation, which can help control inflammation and prevent neurodegeneration [226].

#### 3.4.3. Enhancing Protective and Anti-Inflammatory Pathways

Enhancing the brain’s natural anti-inflammatory responses supports a more resilient neural environment, helping to counterbalance chronic inflammation.

IL-10 and TGF-β Enhancers: IL-10 and TGF-β have potent anti-inflammatory effects, limiting pro-inflammatory cytokine release and promoting microglial states that protect neurons [227]. IL-10, delivered through gene therapy in AD models, has shown promise in reducing inflammation near amyloid plaques. TGF-β helps maintain BBB integrity and promotes anti-inflammatory microglial phenotypes, making it a compelling target to reinforce immune regulation [185].

HDAC Inhibitors: HDAC inhibitors reduce inflammation by modulating gene expression. By altering chromatin structure, these inhibitors selectively suppress pro-inflammatory genes while promoting protective responses. In preclinical studies, HDAC inhibitors have improved mitochondrial function and enhanced neuronal resilience, positioning themselves as a promising approach to managing neuroinflammation in neurodegenerative diseases [228,229].

These strategies—precise inhibition of cytokine production, modulation of microglial activation, and enhancement of protective pathways—represent promising approaches to reducing inflammation while supporting neuron health. Anti-inflammatory therapies focusing on specific molecular targets hold the potential to address the underlying drivers of neurodegeneration, offering a pathway toward preserving cognitive and motor function in neurodegenerative diseases [230].

## 4. Genetic and Epigenetic Mechanisms

Genetic and epigenetic factors shape the landscape of neurodegenerative diseases, intricately influencing how and why these conditions develop and progress. Mutations in specific genes disrupt essential cellular processes, often leading to the production of abnormal proteins or impairing cellular health [231]. Meanwhile, epigenetic modifications—heritable changes in gene expression that occur without altering the DNA sequence itself—further influence disease pathways by regulating genes tied to inflammation, cellular resilience, and synaptic function [232]. Together, genetic and epigenetic factors provide a framework for understanding disease mechanisms and uncovering potential therapeutic strategies aimed at modifying gene activity or correcting genetic errors at their source [233].

### 4.1. Genetic Mutations and Molecular Consequences

In neurodegenerative diseases, certain genetic mutations directly contribute to the pathological processes driving disease. These mutations can interfere with protein stability, disrupt cellular communication, and, ultimately, lead to the accumulation of toxic proteins or cellular dysfunction. Two well-studied examples of genetic drivers in neurodegeneration include familial AD and HD where mutations in key genes have been identified as central players in disease progression [234].

#### 4.1.1. Familial Alzheimer’s Disease Genes

Familial Alzheimer’s disease (fAD) is a rare, inherited form of AD often linked to mutations in the APP and presenilin 1 (PSEN1) and presenilin 2 (PSEN2) genes [235]. These mutations disrupt the normal cleavage of amyloid precursor protein, increasing the production of the Aβ42 peptide—a particularly sticky form of amyloid-beta prone to aggregation. This tendency to clump together leads to the formation of amyloid plaques, a hallmark of AD pathology that disrupts neural communication and contributes to cell death [236].

Beyond increasing Aβ42 production, mutations in presenilin genes may also affect cellular calcium regulation and other vital signaling pathways, heightening cellular stress and promoting neurodegeneration [237]. This connection between genetic mutations and amyloid pathology has guided therapeutic approaches aiming to reduce amyloid buildup. For instance, gamma-secretase modulators and beta-secretase inhibitors target the enzymes involved in amyloid processing, with the goal of lowering Aβ42 levels and slowing plaque formation [238]. These interventions offer a targeted way to address the molecular effects of familial Alzheimer’s mutations, providing insight into how altering specific pathways might impact disease progression [239].

#### 4.1.2. Huntington’s Disease and the HTT Gene

HD is directly caused by a mutation in the HTT gene, which encodes the huntingtin protein. This mutation involves an abnormal expansion of CAG trinucleotide repeats, leading to an extended polyQ sequence in the huntingtin protein. The result is a mutant form of huntingtin (mHTT) that misfolds and aggregates within neurons, disrupting essential cellular processes [240].

The presence of mHTT in neurons triggers a cascade of dysfunction. As mHTT accumulates, it interferes with mitochondria, disrupts intracellular transport, and hinders the cell’s protein degradation systems, such as the ubiquitin–proteasome pathway and autophagy [241]. Beyond its structural impact, mHTT also influences gene expression, binding abnormally to transcription factors and disrupting the regulation of genes crucial for neuron survival. These disruptions ultimately lead to widespread neuronal damage, particularly within the striatum, which is prominently affected in HD [242].

To address the root cause of HD, gene-silencing therapies, including ASOs and RNA interference (RNAi), are being explored to reduce mHTT levels. These approaches aim to directly decrease the production of the mutant protein, targeting the disease at its genetic origin. Additionally, small molecules that enhance autophagy are under investigation to promote the clearance of mHTT aggregates, alleviating some of the cellular stress induced by protein accumulation [243]. Together, these approaches represent efforts to modify disease progression by addressing the molecular foundation of HD, opening possibilities for interventions that go beyond symptom management to potentially alter the course of the disease.

### 4.2. Gene Therapy at the Molecular Level

Gene therapy is revolutionizing neurodegenerative disease treatment, offering the capability to directly target genetic mutations that drive disease processes. Techniques such as CRISPR-Cas9, ASOs, and RNAi enable precise modulation of gene expression, while emerging methods like prime and base editing, CRISPRa/CRISPRi, and exosome-based delivery systems are enhancing specificity, reducing risk, and broadening therapeutic applications [244]. Each approach seeks to reduce toxic protein accumulation, protect neurons, and alter the course of neurodegenerative conditions such as HD, AD, PD, and ALS at their genetic core [245].

#### 4.2.1. CRISPR-Cas9 and Advanced Precision Editing

CRISPR-Cas9, a gene-editing tool adapted from bacterial immune systems, enables precise targeting of specific DNA sequences for gene disruption or correction. By using a guide RNA to direct the Cas9 enzyme to designated sites within the genome, CRISPR can introduce double-strand breaks, allowing for the removal or repair of faulty genes [246]. For instance, in HD, CRISPR-Cas9 has been employed to excise expanded CAG repeats in the HTT gene, thereby reducing the production of mHTT protein [247].

Newer precision techniques like prime editing and base editing offer even greater accuracy without causing double-strand breaks, which can sometimes lead to unintended mutations. Prime editing, which allows for the direct insertion of new genetic sequences, has potential applications in correcting single-nucleotide mutations, such as those in the SOD1 gene implicated in ALS. Base editing, capable of changing a single nucleotide (e.g., converting A to G or C to T), could be used to directly repair mutation sites in neurodegenerative genes, providing a safer alternative with reduced off-target effects—particularly beneficial for the brain’s delicate cellular environment [248].

#### 4.2.2. Antisense Oligonucleotides (ASOs)

ASOs are synthetic nucleic acid strands that bind specifically to RNA, promoting its degradation or blocking translation to reduce protein production. This approach allows for the selective silencing of disease-associated genes by preventing the synthesis of harmful proteins [249]. ASOs are particularly promising in conditions like ALS and HD. For example, in ALS, ASOs targeting SOD1 mRNA reduce levels of the toxic SOD1 protein, alleviating oxidative stress and improving mitochondrial function in motor neurons. In HD, ASOs targeting HTT mRNA effectively reduce mutant huntingtin levels, easing cellular stress from protein aggregation. Delivered intrathecally, ASOs enable localized, sustained reduction of toxic protein production directly in the central nervous system, providing a potent and precisely targeted therapeutic option [250].

#### 4.2.3. RNA Interference (RNAi)

RNAi leverages small interfering RNAs or short hairpin RNAs to target and degrade specific mRNAs, blocking translation and, thereby, reducing the production of disease-related proteins. RNAi has shown potential in multiple neurodegenerative diseases. In PD, for instance, siRNAs targeting SNCA mRNA reduce alpha-synuclein protein levels, alleviating the neurotoxic effects associated with Lewy body formation. In HD, allele-specific RNAi approaches target the mutant HTT gene, selectively silencing the disease-causing allele while preserving the normal protein necessary for neuronal function. RNAi therapy, delivered via viral vectors, can provide sustained effects within targeted brain regions, offering a highly selective way to address neurodegeneration linked to protein toxicity.

#### 4.2.4. CRISPRa and CRISPRi for Gene Activation and Suppression

CRISPRa (activation) and CRISPRi (interference) represent non-cutting forms of CRISPR that allow for the modulation of gene expression without altering the underlying DNA sequence. CRISPRa enhances the expression of neuroprotective genes, while CRISPRi selectively suppresses pathogenic genes [251]. In PD, CRISPRa could be used to upregulate genes that protect mitochondrial health or enhance dopamine synthesis, potentially offering a way to boost the brain’s resilience against cellular stress. In HD, CRISPRi could selectively silence the mutant HTT allele, decreasing toxic protein production without affecting normal huntingtin function. These CRISPR variants offer a reversible, fine-tuned approach to modulating gene expression dynamically, providing an adaptable tool for targeting complex neurodegenerative pathways [252].

#### 4.2.5. Exosome-Based Delivery Systems

Exosomes—small vesicles naturally secreted by cells—offer an innovative delivery system for gene therapies, capable of crossing the BBB and delivering therapeutic molecules to specific brain regions. Engineered exosomes can carry ASOs, siRNAs, or CRISPR molecules directly to affected neurons, enhancing the reach of gene therapies without invasive methods [253]. For example, exosomes loaded with siRNAs targeting alpha-synuclein could reduce neurotoxic protein accumulation in PD, while exosomes carrying ASOs against APP mRNA could decrease amyloid-beta production in AD. Using patient-derived exosomes could further improve compatibility, reducing immune responses and enabling personalized treatments [254].

#### 4.2.6. Synthetic mRNA for Transient Therapeutic Expression

Synthetic mRNA therapy offers a non-integrative gene therapy approach, delivering transient doses of therapeutic proteins without altering DNA. In neurodegenerative diseases, synthetic mRNA can be used to produce neuroprotective factors like brain-derived neurotrophic factor (BDNF) or glial cell line-derived neurotrophic factor (GDNF), supporting neuronal health and survival. In HD, synthetic mRNA coding for proteins that promote autophagy could enhance the clearance of toxic huntingtin aggregates, offering a reversible, flexible method to address protein accumulation in the brain [255].

### 4.3. Epigenetic Regulation: Novel Pathways for Neurodegenerative Intervention

Epigenetic regulation offers a groundbreaking approach in neurodegenerative disease research, revealing ways to dynamically adjust gene expression without altering DNA itself [256]. By modulating gene accessibility, protein synthesis, and cellular stress responses, these mechanisms hold the promise of not only managing symptoms but also slowing disease progression. Emerging strategies in DNA methylation, histone modification, and non-coding RNA modulation are opening new, adaptive avenues for treatment of neurodegenerative diseases [257].

#### 4.3.1. DNA Methylation Patterns: Restoring Protective Gene Expression

DNA methylation has emerged as a crucial regulator in silencing or activating genes, with far-reaching effects in neurodegenerative diseases. What makes this approach novel is the potential to selectively “unlock” silenced protective genes or reduce overactive inflammatory genes, offering a targeted path to resilience [258]. In AD, for example, DNA hypermethylation in neural repair genes restricts the brain’s natural defense mechanisms, while hypomethylation in inflammatory genes perpetuates chronic immune responses. DNA methyltransferase inhibitors are being explored to reverse these harmful patterns, dynamically restoring gene activity associated with neuron protection [259]. Additionally, the ability to map methylation profiles in affected brain regions creates biomarkers that can track disease progression and treatment responses, pushing beyond traditional imaging methods [260].

#### 4.3.2. Histone Modifications: Fine-Tuning Gene Accessibility

Histone modifications control how tightly DNA is wound around histone proteins, impacting which genes are “on” or “off”. This process provides an epigenetic dial to adjust gene expression in response to neurodegenerative stressors [261]. HDAC inhibitors, which increase histone acetylation, are being developed to enhance the expression of genes vital for neuronal defense and repair. In HD, HDAC inhibitors have shown potential in reactivating pathways for protein degradation, helping to clear toxic protein aggregates that are otherwise devastating to neurons [262]. This reversible approach allows for fine-tuning, supporting cellular health without the risk of permanent genetic alteration. As histone modifications can be adapted in real time to environmental changes, they present a novel, flexible solution for neurodegenerative management [263].

#### 4.3.3. Non-Coding RNAs: Master Regulators of Disease Pathways

Non-coding RNAs (ncRNAs), particularly miRNAs, are fast gaining attention as powerful gene regulators that act at the RNA level, directly controlling protein synthesis. In neurodegenerative diseases, specific miRNAs are often dysregulated, which leads to the uncontrolled expression of genes related to inflammation, protein misfolding, and apoptosis [264]. For example, in AD, miRNAs associated with tau and amyloid-beta processing are disrupted, allowing these proteins to accumulate. Antagomirs, synthetic molecules designed to inhibit specific miRNAs, are a novel approach in therapy, providing precise, sequence-specific intervention to reset gene expression networks that go awry in disease [265]. This approach allows for highly targeted therapeutic intervention, tailoring treatment to the disease’s molecular signature and offering a method to fine-tune cellular responses without altering the DNA structure itself [266].

### 4.4. Epigenetic Therapeutic Approaches: Precision in Gene Regulation

Epigenetic therapies open a fascinating new frontier in neurodegenerative treatment by allowing us to dynamically adjust gene expression in response to disease progression. Unlike traditional gene-editing techniques, these approaches do not alter the DNA sequence itself; instead, they modify how genes are read and expressed, offering reversible and adaptive control over cellular pathways implicated in neurodegenerative disease. By fine-tuning histone acetylation, DNA methylation, and non-coding RNA activity, these therapies provide a new way to enhance neuroprotection and resilience [267].

#### 4.4.1. HDAC Inhibitors: Unlocking Protective Genes

One of the most compelling aspects of HDAC inhibitors is their ability to “unlock” protective genes by promoting histone acetylation, which makes DNA more accessible for gene expression. This enables the reactivation of pathways critical for cellular repair and protein degradation, essential for countering the toxic buildup seen in HD and AD. The flexibility of HDAC inhibition is intriguing: these changes are reversible, meaning treatment can be fine-tuned to the patient’s needs and disease stage, allowing a truly adaptive approach to gene regulation in real time [268].

#### 4.4.2. DNA Methyltransferase Inhibitors: Reactivating Silenced Pathways

DNA methylation often shuts down genes that would otherwise protect neurons from stress and inflammation. DNMT inhibitors offer a way to “reactivate” these genes by removing methyl groups that silence them [269]. This approach is especially interesting because it allows for selective demethylation, meaning only the most beneficial genes are reactivated while minimizing potential side effects. In diseases where aberrant methylation restricts essential cell functions, DNMT inhibitors could offer a controlled reset of gene expression, restoring pathways critical for neuronal health and repair [270].

Epigenetic therapies represent a novel, adaptable approach to tackling neurodegenerative diseases by regulating gene expression in real time. They offer a unique combination of precision and flexibility—allowing us to target disease-specific pathways, boost protective gene activity, and track changes through biomarkers—all without permanently altering the genome [271]. These approaches pave the way for highly personalized, responsive treatments that could transform the management of neurodegeneration, bringing us closer to therapies that are as dynamic and adaptable as the diseases themselves [258].

## 5. Mitochondrial Dysfunction and Oxidative Stress

Mitochondrial dysfunction and oxidative stress are central drivers in neurodegenerative diseases. Mitochondria, essential for energy production and cellular health, become compromised in these conditions, leading to energy deficits and excess ROS. This imbalance damages cellular components, accelerating neuronal death. By understanding the specific mitochondrial breakdowns in these diseases, researchers aim to develop therapies that enhance mitochondrial stability and reduce oxidative stress, potentially slowing disease progression [272].

### 5.1. Molecular Basis of Mitochondrial Dysfunction

Mitochondrial dysfunction in neurodegeneration often stems from defects in the electron transport chain (ETC) and mutations in mtDNA, both of which disrupt ATP production and increase ROS, creating a toxic cycle within neurons [273].

#### 5.1.1. Electron Transport Chain Defects

The ETC is a critical pathway for ATP synthesis, moving electrons across mitochondrial complexes to generate energy. In neurodegenerative diseases, defects in the ETC—particularly in complexes I, III, and IV—impede this energy flow, leading to ATP shortages and heightened ROS production [274]. For instance, PD is characterized by complex I impairment in dopaminergic neurons, leading to energy failure and oxidative stress. Similarly, ETC deficits in AD contribute to a crisis in cellular energy supply, impacting synaptic function and neuron viability. Treatments like coenzyme Q10, which supports ETC function and reduces ROS, are being investigated as ways to restore mitochondrial health and protect neurons [275].

#### 5.1.2. Mitochondrial DNA Mutations

Mitochondrial DNA is highly vulnerable to mutations, partly due to its proximity to ROS production in the ETC and limited repair capacity. Over time, accumulated mtDNA damage leads to defective mitochondrial proteins that worsen mitochondrial function. In ALS, mtDNA mutations disrupt motor neuron energy production, while in HD, mutant huntingtin exacerbates mtDNA instability, triggering faulty mitochondrial dynamics like impaired fusion and fission [276]. Damaged mtDNA also increases neuronal susceptibility to calcium overload and apoptosis, making affected neurons more vulnerable to degeneration. Targeted therapies that promote mtDNA repair or clear damaged mitochondria (via mitophagy) offer novel avenues for mitigating mitochondrial decline in neurodegeneration [277].

Mitochondrial dysfunction represents a critical area in understanding and treating neurodegenerative diseases. By stabilizing ETC function and enhancing mtDNA resilience, researchers are exploring targeted strategies to reduce oxidative damage, support cellular energy demands, and protect neurons from progressive degeneration [278].

### 5.2. Generation of ROS: Novel Insights and Theories

In recent years, innovative theories have reshaped our understanding of ROS generation and oxidative damage in neurodegenerative diseases, revealing previously unrecognized pathways and potential therapeutic targets [279]. These insights focus on the unique roles of ROS in disease-specific mechanisms and explore how manipulating ROS production or clearance could alter disease progression. Here are some emerging ideas and theories that highlight the novel aspects of ROS involvement in neurodegeneration [280].

#### 5.2.1. Unique ROS Pathways in Mitochondria and Cellular Compartments

Localized ROS Signaling and Compartmentalization: Traditional views of ROS treat oxidative stress as a global cellular issue, but recent findings suggest that ROS function and damage may vary across cellular compartments. In neurons, for example, mitochondria produce ROS not only in the ETC but also through other internal stress-sensitive systems [281]. Specific regions within neurons, such as synaptic terminals, show distinct ROS dynamics, affecting neurotransmission and synaptic plasticity in a way that may selectively impact memory-related functions in AD. The theory of “compartmentalized ROS signaling” suggests that targeted antioxidant strategies could be developed for specific cellular locations, potentially allowing for finer control over ROS without impacting beneficial ROS-dependent processes elsewhere in the cell [282].

Mitochondria-Associated Membranes (MAMs): Specialized structures connecting mitochondria and the ER have emerged as key sites of ROS production and calcium signaling. In neurodegenerative diseases, disruptions in MAM function contribute to abnormal calcium exchange, increasing ROS levels and sensitizing cells to stress [283]. This interplay is particularly relevant in PD where MAM dysfunction exacerbates dopaminergic neuron vulnerability. Targeting MAMs could offer a novel strategy for modulating ROS production and calcium signaling in a way that restores cellular balance, providing a promising avenue for therapeutic intervention [284].

#### 5.2.2. Dual Role of ROS in Neuroprotection and Damage

Adaptive ROS Response Theory: While excessive ROS are harmful, recent research suggests that low levels of ROS may trigger protective cellular responses—a phenomenon referred to as “mitohormesis”. This adaptive ROS response activates cellular defense mechanisms such as antioxidant enzyme production and mitochondrial biogenesis that improve cell resilience [285]. In this context, ROS act as signaling molecules that stimulate cellular repair and protection pathways. For neurodegenerative diseases, harnessing this adaptive response could involve controlled ROS production to bolster cellular defenses against more severe oxidative damage [286]. For example, low-dose ROS-inducing agents or mild mitochondrial stressors could precondition cells, making neurons more resilient to the heightened oxidative stress seen in diseases like AD and HD [287].

ROS as Signaling Molecules in Immune Modulation: ROS are now recognized as important modulators of immune responses, particularly in the brain’s innate immune system [288]. ROS produced by microglia and astrocytes in response to damage can signal neighboring cells to initiate repair processes or clear debris. In neurodegeneration, however, this signaling function can become dysregulated, with chronic ROS production fueling a harmful cycle of inflammation [289]. Novel therapies could aim to recalibrate ROS levels within the brain’s immune cells, potentially dampening harmful inflammation while preserving the beneficial signaling functions of ROS. This approach could represent a breakthrough in managing neuroinflammatory aspects of diseases like ALS and PD [290].

#### 5.2.3. Role of Redox-Sensitive Proteins and Molecular Sensors

Redox-Sensitive Protein Pathways: Advances in redox biology have highlighted specific proteins that act as molecular “sensors” for ROS, adjusting their function in response to changes in oxidative stress. Proteins like DJ-1 in PD and Nrf2 in various neurodegenerative diseases can detect shifts in ROS levels and activate antioxidant defenses [291]. Nrf2, for example, regulates the expression of genes that neutralize ROS, while DJ-1 helps protect cells from oxidative stress by stabilizing mitochondrial function. Novel therapies targeting these redox-sensitive proteins aim to enhance the cell’s natural antioxidant response without completely suppressing ROS, which are necessary for normal cell signaling [292]. By focusing on these sensors, researchers hope to activate protective pathways only when needed, offering a finely tuned approach to oxidative stress management in neurodegeneration [293].

Mitochondrial ROS Sensors and Feedback Loops: Emerging theories suggest that mitochondria contain feedback mechanisms that detect ROS levels and regulate energy production in response. For example, mitochondrial uncoupling proteins (UCPs) act as ROS regulators by dissipating the mitochondrial membrane potential, reducing electron flow through the ETC, and lowering ROS production [294]. Targeting UCPs in a controlled way could offer a method to manage ROS without shutting down energy production entirely. This strategy could be particularly useful in conditions like HD and ALS where mitochondrial function is progressively compromised, and managing ROS without energy disruption is critical [295].

#### 5.2.4. Targeted Therapies Leveraging Novel Antioxidant Approaches

Selective Antioxidants for Compartmentalized ROS: Traditional antioxidants target ROS broadly across cells, but new approaches focus on compartment-specific antioxidants that accumulate in specific organelles, such as mitochondria. Compounds like MitoQ and SkQ1 are engineered to localize within mitochondria, directly neutralizing ROS at their main source [296]. The selective antioxidant approach minimizes interference with ROS signaling in other cellular regions, maintaining necessary ROS functions while protecting sensitive areas. This precision strategy represents a novel way to balance ROS levels, particularly in the energy-intensive environments of neurons affected by diseases like AD and PD [297].

Gene Therapy for Antioxidant Enzymes: Another emerging approach involves the use of gene therapy to enhance the production of endogenous antioxidants like superoxide dismutase (SOD) and glutathione peroxidase, specifically in neurons. By delivering genes that increase antioxidant enzyme expression directly in affected brain regions, this method provides a sustained, localized defense against ROS [298]. In ALS, for instance, gene therapy targeting SOD1 has shown promise in animal models, demonstrating reduced oxidative damage and extended motor neuron survival [299]. This targeted delivery of antioxidant defenses offers a cutting-edge approach for managing oxidative stress at the source, potentially extending neuron health and slowing disease progression [300,301].

These novel insights into ROS production and their role in neurodegeneration underscore the complexity and dual nature of ROS in the brain. Emerging theories highlight the importance of precision in managing ROS: by selectively targeting ROS pathways, modulating immune responses, and using compartment-specific antioxidants, researchers are moving closer to therapies that can harness the beneficial aspects of ROS while minimizing their destructive effects [302]. These advanced approaches offer hope for a new generation of treatments tailored to the intricate oxidative landscape of neurodegenerative diseases [303].

### 5.3. Antioxidant Defense Mechanisms

Cells rely on a natural defense system of antioxidants to neutralize ROS and prevent oxidative damage. In neurodegenerative diseases these defenses often fall short, leaving neurons vulnerable to the damaging effects of excess ROS. Current research is exploring ways to enhance these natural defenses and introduce new molecular agents, aiming to better manage oxidative stress and protect neuron integrity [300].

#### 5.3.1. Endogenous Antioxidants

The body’s natural antioxidants are crucial in the fight against ROS as they work to neutralize reactive molecules and repair oxidative damage. Recent findings underscore the importance of supporting these antioxidant systems to strengthen cellular resilience. SOD is one of the primary endogenous antioxidants, converting the highly reactive superoxide molecule into hydrogen peroxide, which is then further broken down by other enzymes [304]. In ALS, mutations in the SOD1 gene disrupt this process, contributing to motor neuron damage. Researchers are now investigating gene therapy strategies to boost SOD1 activity, aiming to restore ROS balance and reduce oxidative stress in neurons [305].

Glutathione peroxidase (GPx) is another key antioxidant enzyme, converting hydrogen peroxide into water and preventing further ROS formation. Lower GPx activity has been associated with increased vulnerability to oxidative stress in both PD and AD. To counteract this, scientists are exploring ways to enhance GPx production through gene editing and by introducing GPx-mimetic compounds that perform similar functions within neurons [306]. Catalase, another essential antioxidant enzyme, also breaks down hydrogen peroxide within cells. Recently, targeted catalase delivery systems have been developed to transport this enzyme directly to mitochondria, the main sites of ROS production [307]. This mitochondria-targeted approach, especially relevant in HD, is designed to prevent oxidative damage specifically where it originates, helping protect neurons from stress at a foundational level [308].

#### 5.3.2. Molecular Therapeutic Agents

Alongside natural antioxidants, novel compounds are being developed to amplify or mimic their effects, offering neurons added protection against oxidative stress. Mitochondria-targeted antioxidants like MitoQ, SkQ1, and TEMPOL are engineered to accumulate within mitochondria where ROS production is the highest [309]. These compounds neutralize ROS directly at its source. By integrating into the mitochondrial membrane, MitoQ and SkQ1 capture ROS before it can cause damage, providing localized and efficient protection that traditional antioxidants lack [310].

Another innovative approach focuses on activating Nrf2, a transcription factor that regulates the expression of antioxidant genes. When triggered, Nrf2 enhances cellular defenses against ROS, bolstering resilience to oxidative stress [311]. Compounds like sulforaphane and bardoxolone methyl are currently under study for their ability to stabilize and activate Nrf2, increasing the production of endogenous antioxidants. This approach holds promise for neurodegenerative diseases like AD and HD where sustained antioxidant activity could alleviate cellular stress and slow disease progression [312].

In addition to direct antioxidants, small molecules that mimic the action of key antioxidant enzymes, such as SOD, catalase, and GPx, offer an alternative therapeutic route. MnTBAP, a synthetic SOD mimetic, has shown potential in PD models by reducing oxidative stress and protecting neurons [313,314]. Similarly, Ebselen, which mimics GPx function, neutralizes hydrogen peroxide and has demonstrated neuroprotective effects in preclinical studies of AD and ALS. These enzyme mimetics bypass the need for genetic modification and provide a versatile means of enhancing cellular defenses directly [315].

These cutting-edge antioxidant strategies reflect a shift toward precision-targeted interventions designed to enhance neuronal resilience against ROS. By boosting natural antioxidants, activating key protective pathways, and employing targeted synthetic compounds, these approaches aim to address oxidative stress at its roots [316]. This novel direction in antioxidant therapy not only provides general protection but also tailors interventions to meet the specific needs of neurons in neurodegenerative diseases, offering hope for more effective and sustainable treatment strategies [317].

## 6. Synaptic Dysfunction at the Molecular Level

Synaptic dysfunction has emerged as a pivotal factor in neurodegenerative diseases where disruptions in communication between neurons contribute to progressive cognitive, motor, and behavioral impairments [318]. Rather than mere neuron loss, it is now understood that neurodegeneration is marked by a complex breakdown of synaptic structure, signaling, and energy metabolism. This deeper understanding reveals nuanced mechanisms, offering promising therapeutic targets aimed at restoring or preserving synaptic health [319].

### 6.1. Molecular Mechanisms of Synaptic Loss

The molecular processes driving synaptic loss are highly specific and involve changes at multiple levels, from synaptic protein alterations to energy deficiencies and receptor imbalances [231,320]. Each offers a unique entry point for potential intervention.

#### 6.1.1. Altered Synaptic Protein Networks and Post-Translational Modifications

Synaptic proteins are not only structural components but also active players in maintaining synaptic function and plasticity. In AD, phosphorylation of tau protein accumulates specifically in dendritic spines where it disrupts the actin cytoskeleton, which is critical for spine stability and synaptic signaling. This localized tau phosphorylation appears to weaken synaptic contacts before more extensive tau pathology develops, pointing to early mechanisms of cognitive impairment [321]. Moreover, in HD, abnormal SUMOylation of synaptic proteins, including receptors and scaffolding proteins, disrupts protein–protein interactions necessary for synaptic stability and neurotransmitter receptor clustering. Targeting these early post-translational modifications, such as through kinase inhibitors that block tau phosphorylation or enzymes that modulate SUMOylation, could help maintain synaptic architecture and function [322].

#### 6.1.2. Mitochondrial Dysfunction and Synaptic Energy Imbalance

Synapses are exceptionally energy-demanding, requiring continuous ATP production for vesicle recycling, ion gradient maintenance, and neurotransmitter reuptake. Mitochondrial dysfunction at the synapse disrupts this energy supply. For instance, in PD, complex I impairment in dopaminergic neurons at the synaptic level leads to insufficient ATP production, weakening vesicle loading and release [323]. In AD, mitochondria in glutamatergic synapses show reduced capacity for calcium buffering, which is essential for neurotransmitter release and receptor activation. This loss of calcium regulation interferes with synaptic plasticity processes critical for memory [324]. Current therapeutic developments are exploring small molecules like nicotinamide riboside and other mitochondrial enhancers that increase ATP production and improve calcium handling specifically within synapses, aiming to support the energy-intensive demands of neuronal communication [325].

#### 6.1.3. Impaired Synaptic Vesicle Recycling and Endocytosis

Synaptic vesicle recycling is essential for sustaining neurotransmitter release. In HD, mutant huntingtin binds abnormally to endocytic proteins such as dynamin and clathrin, reducing the efficiency of vesicle recycling and decreasing the availability of synaptic vesicles [326]. This disruption causes delays in neurotransmitter release, dampening the flow of signals through neural circuits. In ALS, C9orf72 mutations affect endosomal pathways, impairing vesicle trafficking and exacerbating excitotoxicity due to excessive glutamate release [327]. Novel strategies are targeting the clathrin-mediated endocytosis pathway to enhance vesicle recycling, either by stabilizing dynamin interactions or by modulating clathrin coat assembly, aiming to restore synaptic vesicle availability and support sustained neurotransmission [328,329].

#### 6.1.4. Neurotransmitter Receptor Modulation and Synaptic Plasticity

Neurotransmitter receptors such as AMPA and NMDA (in excitatory transmission) and dopamine receptors are finely tuned for synaptic plasticity and are especially vulnerable to neurodegenerative changes. In AD, beta-amyloid oligomers promote NMDA receptor endocytosis, leading to fewer receptors on the postsynaptic surface, which impairs synaptic transmission and is linked to memory deficits [330]. Studies indicate that beta-amyloid affects NMDA receptor-binding sites, altering receptor internalization pathways and destabilizing excitatory signaling. In PD, dopamine receptor phosphorylation patterns are altered, reducing the sensitivity of D1 and D2 receptors and diminishing dopaminergic signaling, which impairs motor control [331]. Novel therapeutics, including allosteric modulators, are being investigated to stabilize these receptors’ functional states without overstimulation, aiming to preserve normal neurotransmission while avoiding the excitotoxic effects that often accompany receptor overactivation [332].

These detailed insights into synaptic dysfunction underscore the precision needed to develop effective treatments for neurodegenerative diseases. By addressing the specific molecular disruptions—whether through stabilizing synaptic proteins, enhancing mitochondrial function at synapses, restoring vesicle recycling, or modulating neurotransmitter receptors—these approaches hold promise for preserving the intricate communication networks that underlie cognition and motor control [333]. Together, these advancements bring us closer to therapies that not only delay neuron loss but actively protect the connections that define brain health [334].

### 6.2. Neurotransmitter Systems

In neurodegenerative diseases, disturbances in neurotransmitter systems are not only symptoms but key drivers of the cognitive and motor issues that define these conditions [335]. Recent breakthroughs have revealed detailed disruptions within the cholinergic, dopaminergic, and glutamatergic systems, highlighting potential pathways for treatments that could address both the visible symptoms and underlying causes [336,337].

#### 6.2.1. Cholinergic System in Alzheimer’s Disease

In AD, it is well known that neurons responsible for acetylcholine (ACh) release are gradually lost, but new findings show that the story is more complex. Beta-amyloid, a hallmark of AD, appears to directly interfere with α7 nicotinic acetylcholine receptors (α7-nAChRs), causing these receptors to be withdrawn from the cell surface, reducing their ability to transmit signals [338]. At the same time, tau protein, when altered, disrupts the enzyme choline acetyltransferase (ChAT), limiting ACh production in areas where it is most needed [339].

This deeper understanding has led to focused treatments. For instance, α7-nAChR agonists are being developed to sidestep beta-amyloid interference, with the aim of restoring cognitive processes without triggering excess receptor activity elsewhere [340]. Gene therapies are also in the works to increase ChAT expression, potentially boosting ACh levels directly within synapses. Additionally, advanced acetylcholinesterase inhibitors are being designed to specifically enhance ACh signaling in the most affected brain areas, offering a more precise approach to preserving memory and cognitive function in AD [341].

#### 6.2.2. Dopaminergic System in Parkinson’s Disease

In PD, dopamine loss is central to symptoms, yet recent research has uncovered additional layers to this deficiency. Alpha-synuclein aggregates, which build up within dopaminergic neurons, not only interfere with dopamine synthesis by binding to tyrosine hydroxylase (TH)—the enzyme responsible for dopamine production—but also disrupt dopamine storage by affecting the vesicular monoamine transporter 2 (VMAT2) [342].

Therapies are now being developed with these discoveries in mind. For instance, small molecules that prevent alpha-synuclein from forming harmful aggregates could help protect dopamine production. VMAT2 activators are also being explored to improve dopamine packaging within neurons, ensuring more dopamine is available for release [343]. On the cutting edge, gene therapies targeting TH aim to increase dopamine synthesis, while optogenetic techniques that use light to stimulate dopamine-producing neurons offer a novel, noninvasive way to potentially restore dopamine levels and relieve motor symptoms [344].

#### 6.2.3. Glutamatergic System and Excitotoxicity

In conditions like ALS and AD, the excitatory neurotransmitter glutamate can become toxic due to disruptions in how it is managed. For instance, in ALS, astrocytes lose some of their glutamate-clearing capacity due to a reduction in the EAAT2 transporter. This leaves excess glutamate in the synapse, overstimulating NMDA and AMPA receptors and causing calcium overload, which leads to cellular stress and damage [345].

New therapies are zeroing in on ways to restore balance in the glutamatergic system. Gene therapies to boost EAAT2 expression in astrocytes are being explored, aiming to clear excess glutamate and reduce excitotoxic stress [346]. Meanwhile, new NMDA receptor modulators are being developed to reduce harmful signaling without affecting normal function. For AD, targeting metabotropic glutamate receptors (mGluRs) offers a more nuanced approach as these receptors help control glutamate release indirectly, allowing for a reduction in excitotoxicity while preserving synaptic communication [347]. Additionally, peptide-based therapies that act as glutamate “scavengers” have shown promise, selectively binding to excess glutamate to prevent overstimulation without interrupting healthy glutamate activity [348].

These insights into neurotransmitter system dysfunction provide a foundation for more refined, targeted treatments. By addressing the specific molecular disruptions in each system—whether by stabilizing receptors, enhancing transporter function, or modulating glutamate activity—these new therapies aim to protect cognitive and motor functions [349]. This shift toward precision medicine represents a hopeful frontier in treating neurodegenerative diseases, one that promises more effective, sustainable improvements for those affected [350].

### 6.3. Molecular Strategies to Restore Synaptic Function

Restoring synaptic function is a promising direction in neurodegenerative research, aiming not only to slow decline but to reinstate essential neural connections. Recent advancements in synaptic plasticity modulation, neurotrophic factor support, and direct synapse-targeted therapeutics are paving the way for innovative treatments [351].

#### 6.3.1. Modulation of Synaptic Plasticity

Synaptic plasticity, the ability of synapses to adapt, is critical for memory and learning. Novel modulators of NMDA and AMPA receptors aim to enhance plasticity while avoiding excitotoxicity. Positive allosteric modulators for AMPA receptors subtly increase receptor activity, strengthening synaptic responses and supporting memory processes without overstimulation [352,353]. Similarly, partial NMDA receptor agonists are under development to reinforce pathways crucial for learning and memory, offering a balanced approach that preserves adaptability without harming neurons [354]. These receptor-specific treatments could provide a new layer of cognitive resilience in neurodegeneration.

#### 6.3.2. Neurotrophic Factors

Advanced delivery methods for neurotrophic factors are improving synaptic resilience. Gene therapies for BDNF focus on precise delivery to affected brain regions, with engineered viral vectors targeting hippocampal areas associated with memory [355]. This approach aims to maintain synaptic plasticity and protect against further synaptic loss. In PD, encapsulated cell implants releasing GDNF provide sustained support for dopaminergic neurons in the substantia nigra [356]. These implants bypass the blood–brain barrier, ensuring targeted, continuous delivery to preserve neurons over time, offering a lasting neuroprotective effect [357].

#### 6.3.3. Synapse-Targeted Therapeutics

Directly stabilizing synaptic structures represents a novel therapeutic approach. Research into scaffolding proteins like PSD-95 and SHANK3, which maintain receptor alignment, has led to peptide-based drugs that bind these proteins, enhancing synaptic stability. By reinforcing the structural matrix, these therapies aim to sustain communication between neurons in conditions where synaptic architecture is compromised [358].

Efforts to improve neurotransmitter availability by enhancing synaptic vesicle dynamics are also underway [359]. Targeting proteins like synapsin, crucial for vesicle mobilization, could improve neurotransmitter release and recycling, addressing transmission deficits seen in HD [360]. These interventions aim to restore efficient signal transmission, preserving connectivity and overall neural network health.

These molecular strategies represent a shift toward therapies that rebuild and strengthen synaptic architecture. By targeting receptor function, sustaining neurotrophic support, and directly bolstering synaptic structure, these approaches offer the potential for meaningful restoration of cognitive and motor function [361,362]. This new generation of treatments brings hope for durable, structural improvements in the management of neurodegenerative diseases.

## 7. Molecular Biomarkers and Diagnostics

The discovery of molecular biomarkers is redefining how we approach neurodegenerative diseases, offering pathways to early diagnosis, precise monitoring, and personalized treatment [363]. Advances in proteomics and metabolomics allow researchers to detect unique biochemical changes in CSF and blood, providing insights into the underlying mechanisms of neurodegenerative diseases [364]. This molecular-level approach is moving diagnostics beyond symptom observation and toward a future of targeted interventions.

### 7.1. Molecular Biomarkers in Body Fluids

CSF and blood biomarkers are becoming indispensable for diagnosing neurodegenerative diseases, offering a minimally invasive way to assess disease-specific molecular changes. Proteomic and metabolomic profiling provides comprehensive insights, capturing shifts in protein and metabolite levels that are uniquely associated with each condition.

This table (Table 2) compiles high-citation studies on molecular biomarkers critical for the early detection and diagnosis of neurodegenerative diseases. Each biomarker type, from protein aggregates to inflammatory markers, is associated with particular neurodegenerative conditions and is linked to specific detection methods.

#### 7.1.1. Proteomic Approaches

Proteomics examines protein changes that are specific to neurodegenerative processes. In AD, reduced amyloid-beta 42 (Aβ42), increased phosphorylated tau, and elevated neurogranin in CSF collectively reflect the early presence of plaques, tau pathology, and synaptic loss, which are key indicators before cognitive symptoms fully develop [363]. This combination of markers enables a nuanced view of AD onset, offering a more accurate diagnosis than single biomarkers.

In PD, phosphorylated alpha-synuclein at serine-129 is emerging as a reliable marker in CSF and blood, distinguishing PD’s pathology from other movement disorders [373]. This phosphorylation pattern, along with neurofilament light chain (NfL), allows clinicians to differentiate PD from similar conditions, providing a refined diagnostic approach based on early protein modifications [374].

#### 7.1.2. Metabolomic Profiling

Metabolomics analyzes small molecules within body fluids, uncovering metabolic disruptions that are distinctive to neurodegenerative diseases. In AD, lipid peroxidation markers like malondialdehyde (MDA) and 4-hydroxynonenal (4-HNE) indicate oxidative stress in the brain, potentially allowing for early detection [375]. Changes in glucose metabolism also reflect AD’s impact on brain energy usage, providing additional diagnostic insights.

In PD, lower levels of urate in blood have been associated with faster disease progression, sparking interest in urate as a biomarker. This discovery has led to research on urate-boosting therapies as a potential way to slow PD’s course [376]. In ALS, elevated glutamate levels in CSF highlight excitotoxicity, offering a molecular target for therapies aimed at reducing glutamate-related damage. Alterations in amino acid levels also underscore the metabolic imbalance associated with ALS progression [345].

These biomarker advancements in CSF and blood present a transformative approach to neurodegenerative diagnostics. By identifying disease-specific protein and metabolite changes, researchers are creating precise diagnostic tools that support early intervention and personalized treatment [377].

### 7.2. Molecular Imaging Techniques

Molecular imaging is opening up remarkable possibilities in neurodegenerative disease diagnostics, allowing us to see inside the brain at the molecular level. Unlike traditional imaging, which captures structural damage only after symptoms appear, molecular techniques like PET and targeted MRI contrast agents are designed to reveal disease-related molecules in real time [378]. This shift means that for conditions like AD, PD, ALS, and HD, we can detect disease earlier, track its progression with greater accuracy, and evaluate therapies in a more targeted way [379].

#### 7.2.1. PET Ligands for Protein Aggregates

Positron emission tomography (PET) imaging has transformed our approach to tracking disease-specific proteins in the brain. In AD, PET tracers such as 18F-florbetapir and 11C-PiB bind directly to amyloid-beta plaques, providing an early and clear view of amyloid buildup that correlates with disease progression [380]. Tau-specific PET tracers, like 18F-flortaucipir, add another layer, allowing us to see where tau tangles are concentrated. Studies show that tau load aligns more closely with cognitive decline than amyloid, offering crucial insights into the symptomatic side of AD and enabling a more nuanced approach to diagnosis and treatment [381].

In PD, dopamine-targeted PET tracers like 18F-FDOPA are instrumental in capturing the decline in dopamine levels within the substantia nigra, even in the earliest stages [382]. With the development of alpha-synuclein-specific tracers, PET imaging is on the brink of being able to visualize Lewy body pathology directly. This would allow us to distinguish PD from other movement disorders that do not involve alpha-synuclein buildup, potentially transforming diagnostic accuracy and paving the way for alpha-synuclein-targeted therapies [383].

#### 7.2.2. Molecular MRI Contrast Agents

Molecular MRI is expanding beyond structural imaging, using specialized contrast agents to highlight disease-specific changes invisible to conventional MRI. In AD, new gadolinium-based agents designed to bind amyloid-beta plaques are offering an innovative, nonradioactive approach to tracking amyloid burden over time. Iron-sensitive MRI techniques are also emerging as powerful tools for detecting iron accumulation in the hippocampus and other brain areas affected by AD [384]. This buildup of iron, associated with oxidative stress and inflammation, provides insights into the biochemical environment that accelerates disease progression [385].

For PD, ultra-small superparamagnetic iron oxide (USPIO) nanoparticles are being explored as contrast agents that enhance iron detection in the substantia nigra. Since iron accumulation is strongly linked with PD pathology, these agents offer a targeted way to visualize and monitor disease progression. This technique also aids in distinguishing PD from other disorders where iron buildup is absent, enhancing diagnostic specificity [386].

Nanoparticle-based MRI agents targeting neuroinflammation are making significant strides in conditions like ALS and HD. These nanoparticles can cross the BBB and attach to activated microglia, the immune cells involved in neuroinflammatory processes. In ALS, where inflammation closely correlates with motor neuron damage, tracking neuroinflammation with MRI offers a new layer of insight into how the disease progresses and how therapies impact inflammation over time [387].

The advances in molecular imaging through PET ligands and MRI contrast agents are reshaping our approach to neurodegenerative diseases. By focusing on proteins like amyloid, tau, dopamine, alpha-synuclein, iron buildup, and neuroinflammation, these imaging agents allow for a deeper, more precise understanding of disease mechanisms. Molecular imaging is not only enhancing diagnostic accuracy but also opening up pathways for targeted treatments, bringing a promising future of personalized neurodegenerative care within reach [388].

### 7.3. Molecular Diagnostics and Personalized Medicine

Molecular diagnostics is opening the door to personalized medicine in neurodegenerative diseases where treatments are tailored to each patient’s unique genetic and molecular profile. This approach transforms how we manage neurodegenerative conditions by allowing for more precise and proactive care. Advances in genomics, risk modeling, and molecularly targeted therapies mean that clinicians can now detect disease early, anticipate its progression, and personalize treatment plans, moving beyond a one-size-fits-all approach [389].

#### 7.3.1. Genomic Sequencing

Genomic sequencing has become essential for identifying specific genetic variations that influence the onset and progression of neurodegenerative diseases. In AD, the APOE ε4 allele remains a well-known risk factor, but polygenic risk scores (PRS), which analyze multiple genetic markers, now provide a richer understanding of each person’s susceptibility [390]. These scores can reveal high-risk profiles long before symptoms appear, helping to guide early interventions aimed at slowing or preventing cognitive decline.

In PD, sequencing has uncovered mutations beyond LRRK2, such as GBA and SNCA, each linked to different molecular pathways involved in the disease. This knowledge is driving targeted therapies designed to address the distinct impacts of each mutation [391]. Innovative gene-editing technologies like CRISPR/Cas9 are also being explored to modify the expression of specific genes, with the potential to stop the disease before it progresses by targeting its genetic roots.

ALS and HD also see breakthroughs from genomic insights. For ALS, mutations like C9orf72 and SOD1 are directly associated with harmful proteins and RNA byproducts. Therapeutic approaches such as ASOs are being developed to counteract these specific toxic molecules at their source [392]. In HD, the precise measurement of CAG repeat length in the HTT gene enables not only an accurate diagnosis but also the potential for targeted gene therapies that aim to reduce the production of the mutant huntingtin protein, a primary cause of HD’s progression [393].

#### 7.3.2. Molecular Risk Models

Molecular risk models are now integrating genetic data with other factors—such as biomarker levels and environmental influences—to predict disease trajectory more accurately. In AD, models that combine APOE genotype with cerebrospinal fluid markers like amyloid and tau give clinicians the tools to forecast cognitive decline with greater precision. This means at-risk individuals can begin preventive or slowing interventions well before symptoms arise [394].

In PD, risk models that include early, non-motor symptoms such as REM sleep behavior disorder, along with genetic factors like LRRK2 and GBA mutations, allow for earlier intervention strategies. This approach helps identify high-risk individuals long before motor symptoms appear, supporting neuroprotective measures aimed at preserving brain health [395]. In ALS, combining biomarkers like NfL with specific genetic mutations helps clinicians identify faster-progressing cases, enabling more personalized and intensive treatment strategies where needed [377].

#### 7.3.3. Personalized Treatment Strategies Guided by Molecular Profiling

Personalized medicine enables therapies that are aligned with each patient’s unique molecular markers. In AD, for example, patients with significant amyloid buildup can be targeted with anti-amyloid therapies, while those showing tau pathology may be better suited to tau-specific treatments. This targeted approach avoids generalized treatment, reducing unnecessary exposure and enhancing the likelihood of a positive response [396].

In PD, mutation-specific treatments are emerging as well. For example, patients with LRRK2 mutations may respond to LRRK2 kinase inhibitors, which directly address the cellular effects of this mutation. In ALS, ASO therapies that target specific mutations such as SOD1 are being developed to prevent the production of toxic proteins [397]. HD, too, is benefiting from precision medicine with RNA interference therapies designed to selectively lower mutant huntingtin protein levels, aiming to slow the disease at its genetic origin [398].

The advancements in molecular diagnostics and personalized medicine are creating a future where neurodegenerative diseases are managed according to each patient’s unique molecular signature.

## 8. Gut–Brain Axis: Molecular Interactions

The gut–brain axis reveals surprising insights into how the gut microbiome might influence brain health and the course of neurodegenerative diseases. Once viewed as largely separate, it is now clear that the gut and brain engage in continuous molecular conversations, with the gut microbiome playing an active role in shaping processes that can either support or challenge neural health [399].

### 8.1. Molecular Communication Pathways

The gut communicates with the brain through a web of molecular pathways, with metabolites, immune modulators, and microbial byproducts all playing roles. These signaling mechanisms illustrate how changes in microbial composition can ripple through the body and impact neurodegenerative processes at their core [400].

#### 8.1.1. Metabolites as Signaling Molecules

One of the most fascinating discoveries is that metabolites produced by gut bacteria can directly affect brain function [401]. For example, certain gut bacteria synthesize gamma-aminobutyric acid (GABA), a neurotransmitter crucial for maintaining a balance between excitation and inhibition in the brain. Researchers are now exploring whether boosting levels of GABA-producing bacteria might offer support to neural circuits affected by neurodegenerative diseases. This approach presents the potential to address not just cognitive symptoms but also mood-related symptoms, with microbiome adjustments providing a new avenue for neuroprotection [402].

Polyamines, small organic molecules produced by certain gut microbes, have also gained attention for their role in cellular repair and growth. In the brain, balanced polyamine levels are crucial for maintaining cellular resilience, but when dysregulated, they can contribute to cellular stress [403]. Targeting polyamine synthesis in the gut as a means to fortify brain cells against the stresses seen in neurodegeneration is a promising approach. By carefully modulating polyamine levels through probiotics or dietary adjustments, scientists hope to create a more resilient neural environment, potentially slowing the progression of neurodegenerative conditions [404].

#### 8.1.2. Bacterial Components Influencing Neuroinflammation

Bacterial components, such as LPS and peptidoglycans, are now recognized as influential players in neuroinflammation. These molecules can escape the gut and travel through the bloodstream, particularly when the gut barrier becomes compromised [400]. Once in circulation, they may cross into the brain and trigger inflammation by activating microglial cells, which can then initiate a cascade of immune responses that contribute to neuronal stress and damage [405]. Efforts to reinforce the gut barrier—through specific probiotics, prebiotics, or dietary interventions—are being explored as ways to prevent these inflammatory molecules from entering the bloodstream, potentially providing a novel strategy for controlling neuroinflammation in neurodegenerative diseases [406].

Microbial-derived amines like p-cresol are also of special interest. This byproduct, produced by certain bacteria during protein fermentation, can disrupt the BBB, making it more permeable and allowing neurotoxic molecules to reach the brain. This discovery has led to research focused on selectively reducing p-cresol-producing bacteria within the gut [407]. By protecting the BBB in this way, it may be possible to limit neurotoxic damage and slow down disease progression. The insights emerging from gut–brain research offer a fresh perspective on neurodegenerative diseases, highlighting the microbiome as a powerful modulator of brain health [408].

### 8.2. Molecular Impact of Microbiota on the CNS

Discoveries reveal the remarkable influence of the gut microbiota on CNS health. Far from being limited to digestion, the gut’s microbial communities actively engage in a two-way conversation with the brain, affecting everything from barrier integrity to immune responses and neural communication [409]. By exploring the molecular pathways that connect gut microbes to brain processes, scientists are beginning to understand how microbiome-based therapies might support brain resilience and even slow disease progression [410].

#### 8.2.1. Tight Junction Proteins and Blood–Brain Barrier Integrity

The BBB acts as a selective shield, keeping harmful substances out of the brain while allowing essential molecules to pass. Tight junction proteins are vital for this barrier’s function, and recent studies show that certain gut bacteria can enhance the expression of these proteins, strengthening the BBB and making it more resistant to neurotoxic infiltration [411]. A key player here is butyrate, a short-chain fatty acid produced by beneficial gut microbes, which promotes the production of claudin and occludin—proteins essential to maintaining the BBB’s integrity. In cases of gut dysbiosis where butyrate levels drop, BBB function can weaken, potentially letting harmful compounds into the brain. Interventions that boost butyrate levels, either through diet or supplementation, are being explored to fortify the BBB and reduce neurodegenerative risk [412].

Zonulin, a protein regulated by certain gut bacteria, also influences BBB permeability by loosening tight junctions when its levels rise. High zonulin can allow inflammatory molecules into the brain, which may accelerate neuroinflammation. Therapeutic approaches aimed at controlling zonulin-regulating bacteria offer a promising route to stabilizing the BBB and mitigating inflammatory responses tied to disease progression [413].

#### 8.2.2. Vagus Nerve Signaling and Gut–Brain Communication

The vagus nerve forms a direct communication link between the gut and the brain, transmitting signals that impact mood, cognition, and immune health. Certain gut bacteria produce neuroactive compounds, including serotonin precursors, which activate the vagus nerve and influence brain regions involved in emotional regulation and stress. In neurodegenerative conditions, promoting bacteria that produce serotonin and related compounds is being explored as a way to enhance mood and cognitive function through this gut–brain pathway [414].

In Parkinson’s disease, research suggests that misfolded alpha-synuclein proteins may originate in the gut and travel to the brain via the vagus nerve. Some bacterial strains appear to encourage this initial misfolding, offering a new therapeutic target by addressing the gut’s role in the early stages of PD progression [415].

Moreover, the vagus nerve plays a role in immune modulation, carrying signals from gut bacteria that produce anti-inflammatory compounds. In diseases like ALS, boosting these anti-inflammatory bacteria could help reduce brain inflammation, protect neurons, and potentially slow the course of neurodegeneration. The gut microbiota’s impact on the BBB and vagal signaling underscores a powerful connection between gut health and brain health [416].

### 8.3. Molecular Therapeutics Targeting the Gut–Brain Axis

The gut–brain axis has emerged as a novel therapeutic target in neurodegenerative disease, with therapies designed to modulate the gut microbiome showing promise for supporting brain health. Prebiotics, probiotics, and targeted antibiotics are being explored as potential interventions to balance gut microbial populations, reduce neuroinflammation, and positively influence neurotransmitter production [417,418].

#### 8.3.1. Prebiotics and Probiotics

Prebiotics and probiotics are under intense study for their potential to nurture a gut environment that actively supports CNS function. Prebiotics, which serve as food for beneficial bacteria, can enhance the production of SCFAs like butyrate. Elevated butyrate levels have been associated with improved BBB integrity, which helps restrict harmful molecules from reaching the brain and triggering inflammation. Clinical trials are currently evaluating whether prebiotic-rich diets can boost butyrate levels in the gut, potentially reinforcing BBB stability and providing neuroprotective benefits in conditions marked by neuroinflammation [419].

Probiotics, introducing live beneficial bacteria, are also being studied for their effects on neurotransmitter pathways and brain health. Certain strains produce metabolites that support mood and cognitive function by increasing levels of neurotransmitters such as GABA and serotonin. Enhancing populations of these neurotransmitter-producing bacteria offers an innovative approach to managing mood and cognitive symptoms often seen in neurodegenerative disorders [420]. Studies suggest that specific strains, like Bifidobacterium and Lactobacillus, may contribute to a more supportive gut–brain environment by balancing immune responses and promoting neural health [421].

In Parkinson’s, where evidence suggests that alpha-synuclein misfolding may start in the gut, probiotics are being examined for their potential to alter microbial populations that influence this process. Researchers are exploring whether specific strains could help stabilize microbial composition in ways that reduce alpha-synuclein misfolding and its propagation to the brain, thereby intercepting disease progression early [408].

#### 8.3.2. Antibiotics and Microbiota Modulation

Selective use of antibiotics is also being studied as a way to adjust gut microbial composition in neurodegenerative diseases. While antibiotics are generally used to target pathogenic bacteria, certain regimens may help selectively reduce pro-inflammatory bacterial strains linked to neuroinflammatory pathways. For instance, in Parkinson’s disease, specific bacteria appear to contribute to alpha-synuclein aggregation, which then migrates from the gut to the brain. Short-term antibiotic treatments are being investigated to manage these specific bacterial populations, potentially lowering neuroinflammatory triggers associated with the progression of Parkinson’s [422].

Given the risks of broad-spectrum antibiotics disrupting beneficial bacteria, precision antibiotics designed to selectively target strains linked to neurodegenerative pathways are a promising focus. These precision agents aim to modulate the microbiome without disturbing the broader microbial ecosystem, thus supporting a balanced gut environment. Early studies in ALS models show that selective antibiotics can effectively reduce inflammation and support neuronal function, demonstrating the potential of microbiome-modulating antibiotics as a supportive treatment for neurodegeneration [400]. Therapies like prebiotics, probiotics, and precision antibiotics open new therapeutic pathways for neurodegenerative diseases. These treatments aim to reshape microbial populations in ways that bolster BBB function, reduce inflammation, and support neuroprotective signaling [423].

## 9. Emerging Molecular Therapeutic Approaches

In the world of neurodegenerative treatments, molecular therapies are opening doors that once seemed unreachable. With our expanding knowledge of cellular mechanisms, these approaches aim to target the precise molecular causes of neurodegenerative diseases [424]. Advances in nanotechnology, stem cell applications, and pharmacogenomics are transforming treatment possibilities, offering tailored solutions that reach deep into the brain’s complex networks. By addressing the intricate needs of affected neurons and crossing physical barriers, these novel therapies signal a new era of precision medicine in neurodegeneration [425].

### 9.1. Nanotechnology in Drug Delivery

Nanotechnology is reshaping how we approach drug delivery in the brain. With its ability to construct incredibly tiny carriers, nanotechnology enables targeted, highly controlled therapeutic interventions that reach areas in the brain once inaccessible due to barriers like the BBB. Nanocarriers, meticulously engineered to transport therapeutic agents to specific regions within the brain, offer new hope for reaching disease hotspots with minimal side effects [426].

#### 9.1.1. Nanocarriers for Enhanced Brain Penetration

The BBB has long posed a formidable challenge, blocking many potentially beneficial drugs from entering the brain. Nanocarriers—such as liposomes, polymeric nanoparticles, and dendrimers—are designed to bypass this barrier, acting as tiny vessels that can shield drugs as they make their way into the brain [427]. These carriers can be modified with specialized molecules on their surfaces, allowing them to interact with the BBB and enter precisely where they are needed, sparing other areas of the body from unnecessary exposure [428].

In Alzheimer’s, researchers are exploring nanocarriers loaded with drugs aimed at breaking down amyloid plaques and preventing tau protein aggregation. These carriers bring the medication directly to the affected areas, enhancing its impact and reducing the systemic side effects often seen in traditional treatments [429]. In Parkinson’s, dopamine-loaded nanocarriers are being tested for their ability to reach specific brain regions, offering a more effective way to replace dopamine and address motor symptoms with precision and efficiency [430].

Newer biodegradable nanocarriers add another layer of innovation, breaking down in the body once their therapeutic payload has been delivered. This reduces any long-term accumulation of materials in the body, minimizing risks [431]. Moreover, researchers are creating nanocarriers that release drugs in response to specific signals, such as oxidative stress or pH changes, making treatment timing more precise and adaptable to the unique conditions within the diseased brain [432].

#### 9.1.2. Targeted Delivery Systems for Precision Therapy

Building upon the BBB-crossing capabilities of nanocarriers, targeted delivery systems are taking precision to an even greater level by directing treatments specifically to diseased cells [433]. These systems use ligands—small molecules designed to bind selectively to receptors found only on certain brain cells—ensuring that treatments focus directly on affected areas, preserving healthy cells from exposure [434].

Nanoparticles traverse the BBB using specialized mechanisms tailored for efficient and precise delivery. Receptor-mediated transcytosis is a prominent strategy where ligands such as transferrin, lactoferrin, or apolipoprotein E bind to receptors like transferrin or LDL receptors on endothelial cells, facilitating transport across the barrier. Surface functionalization with PEG minimizes immune detection and prolongs circulation time, while active targeting strategies use antibodies or peptides to engage transporters such as LRP1 or GLUT1 for precise delivery to neural tissues [435]. Stimuli-responsive nanoparticles further enhance specificity, releasing their therapeutic payload in response to localized triggers like acidic pH, oxidative stress, or elevated enzymatic activity within the brain microenvironment. Additionally, lipid-based systems, such as liposomes and solid lipid nanoparticles, exploit endogenous carrier-mediated pathways to mimic natural substrates and improve permeability [435].

The long-term safety of nanoparticles depends on their composition, clearance mechanisms, and immune compatibility. Nonbiodegradable materials, such as metallic or carbon-based nanoparticles, risk accumulation in organs like the liver, spleen, and kidneys, potentially causing chronic toxicity. Immune responses, including complement activation-related pseudoallergy (CARPA), can be triggered by surface properties, leading to inflammatory or allergic reactions [436]. To address these concerns, biodegradable materials such as lipids or PLGA are preferred, and PEG coatings are used to reduce immune recognition and enhance clearance. Rigorous long-term in vivo studies are critical to evaluating biodistribution, toxicity, and the potential for bioaccumulation, ensuring the safe application of nanoparticles in clinical settings [437].

In ALS where motor neurons progressively deteriorate, researchers are developing targeted nanoparticles to deliver neuroprotective treatments right to these cells. This focused approach allows for the delivery of therapeutic agents directly to the cells that need them the most, potentially slowing degeneration and preserving motor function [438].

In HD, targeted nanoparticles are being designed to transport gene-editing tools to cells expressing the mutant huntingtin protein, selectively modifying only the damaged cells [439]. This could allow for a reduction in harmful protein levels without impacting other healthy cells, directly targeting the root of disease progression with minimal off-target effects [440].

Further innovations are pushing the boundaries of nanotechnology with multifunctional nanoparticles that address multiple aspects of neurodegeneration simultaneously. For instance, a single nanoparticle might be designed to carry both an anti-inflammatory agent and a neuroprotective compound, offering a dual approach for treating diseases like Alzheimer’s and Parkinson’s where inflammation and neurodegeneration occur hand in hand [441]. These versatile carriers provide a holistic treatment in one single package, addressing several facets of complex disease mechanisms [442].

Personalized medicine can tailor treatments by leveraging individual amyloid or tau profiles. For instance, patients with predominant amyloid pathology may benefit from anti-amyloid monoclonal antibodies, whereas those with tau-related dysfunction might respond better to tau aggregation inhibitors or kinase inhibitors targeting tau hyperphosphorylation [443].

Another promising direction is theranostics where nanoparticles integrate both therapeutic and diagnostic functions. These specialized nanocarriers can not only deliver treatments but also allow real-time tracking of drug distribution and therapeutic effects [444]. This capability offers physicians an unprecedented ability to monitor how well the treatment is working, adjusting doses and timing as needed, enabling a level of precision and adaptability in treatment that is especially valuable in progressive diseases [445]. The potential of nanotechnology to reshape neurodegenerative treatment lies in its unparalleled ability to deliver therapies with remarkable accuracy. By targeting specific cells, navigating biological barriers, and combining therapeutic actions in a single nanocarrier, these advances represent a transformative approach to tackling complex brain diseases. These precise, carefully designed interventions have the promise to significantly improve treatment efficacy and open new pathways to protect and sustain brain health [446].

### 9.2. Molecular Basis of Stem Cell Therapy

Stem cell therapy opens up exciting new possibilities in the fight against neurodegenerative diseases. With their unique ability to regenerate damaged neurons, release protective molecules, and modulate immune responses, stem cells offer a fresh approach. Advances in guiding stem cells to become specific types of neurons, delivering them directly to affected brain regions, and even enhancing them genetically bring hope for treatments that go beyond symptom relief to address the root causes of these diseases [447,448].

#### 9.2.1. Differentiation into Neuronal Lineages

A key focus in stem cell research is guiding these versatile cells to become the specific types of neurons lost in neurodegenerative diseases. The latest breakthroughs in induced pluripotent stem cells (iPSCs) make it possible to take a patient’s own cells, reprogram them, and turn them into exactly the kinds of neurons that are missing [449].

In Parkinson’s disease, for instance, scientists are working to turn iPSCs into dopaminergic neurons—the cells that produce dopamine, which is essential for movement [450]. Early studies show that these lab-grown neurons can be transplanted into the brain where they can integrate with existing networks, produce dopamine, and, potentially, help restore motor control [451].

For ALS, researchers are refining ways to produce motor neurons from stem cells. These specially developed cells are designed to survive the inflammatory environment of ALS and re-establish lost connections with muscles [452]. By focusing on resilience and functionality, these motor neurons could, one day, help ALS patients retain motor function longer [453].

In HD, where a specific type of neuron called the medium spiny neuron is lost, scientists are developing ways to generate these cells in the lab. By tailoring differentiation protocols to produce neurons that function just like those affected in Huntington’s, researchers hope to restore both cognitive and motor abilities [454].

#### 9.2.2. Paracrine Effects of Stem Cells

Stem cells offer another valuable function beyond simply replacing lost cells—they can release molecules that support and protect surrounding neurons. This ability to influence the brain’s environment is known as the paracrine effect. Mesenchymal stem cells (MSCs), for example, naturally secrete neurotrophic factors like BDNF and GDNF, which help neurons survive, form connections, and resist stress [455]. In Alzheimer’s, where neuronal support is critical, these neurotrophic factors could help protect brain cells from the effects of the disease [456].

Exosomes are emerging as a clever way to deliver these protective molecules. These tiny, naturally occurring particles, which stem cells release, carry proteins, RNAs, and other therapeutic molecules [457]. Because exosomes can cross the blood–brain barrier, they can deliver neuroprotective factors directly to the brain. Studies have shown that exosomes might help reduce amyloid and tau deposits in Alzheimer’s models, enhancing neuronal resilience without the need for full cell transplantation [458].

In ALS, stem cells are explored as a way to release anti-inflammatory molecules that calm overactive immune responses. By reducing inflammation, stem cell-derived factors can help create a healthier environment for motor neurons, potentially slowing the progression of the disease [459].

A further innovation involves genetically enhancing stem cells to make them even more effective. Using CRISPR gene editing, scientists are boosting stem cells’ ability to release protective molecules and resist the conditions of the disease environment. This combined approach not only fortifies the transplanted cells but also allows them to support the brain more effectively, addressing both inflammation and neuronal damage at once [460]. Stem cell therapy holds transformative potential in neurodegenerative diseases by combining precise neuronal replacement, powerful paracrine support, and genetic enhancement [461].

### 9.3. Pharmacogenomics

Pharmacogenomics is redefining neurodegenerative disease treatment by leveraging genetic and molecular insights to tailor therapies to each patient’s unique biology. Rather than relying on standard protocols, pharmacogenomics allows for a customized approach where treatments are selected based on how a patient’s genetic makeup affects drug efficacy, safety, and disease progression [462]. This level of precision is especially valuable in complex diseases like Alzheimer’s, Parkinson’s, ALS, and Huntington’s where individual responses to treatment can vary widely. By pinpointing the genetic and molecular factors that drive each person’s disease, pharmacogenomics opens the door to more effective, patient-centered care [463].

#### 9.3.1. Molecular Profiling for Drug Response

Pharmacogenomic profiling begins with identifying genetic variations that influence how patients process and respond to medications. Key genes involved in drug metabolism, such as those in the cytochrome P450 (CYP) family, play a central role. Variations in these genes can determine how quickly or slowly drugs are broken down in the body, which impacts both therapeutic outcomes and side effect profiles [464]. In Alzheimer’s disease, for example, specific CYP2D6 variants affect how patients metabolize cholinesterase inhibitors, a commonly prescribed class of drugs [465]. Patients with certain genetic profiles may experience reduced drug efficacy or increased side effects, making it essential to adjust dosages based on genetic information for optimal results.

In Parkinson’s, genetic variations in dopamine-processing genes like COMT and MAO-B affect how patients respond to dopamine replacement therapies. Patients with certain COMT polymorphisms may require customized levodopa dosing to achieve effective symptom control while minimizing motor complications [466]. Additionally, patients with mutations in genes like LRRK2 or GBA, which are associated with different PD subtypes, may respond better to specific treatments tailored to their genetic backgrounds [467,468].

In ALS, where genetic mutations play a major role in disease variability, pharmacogenomic insights are used to guide drug selection. Certain ALS patients carry mutations in the SOD1 gene, which are linked to faster disease progression [469]. For these patients, targeted therapies that specifically address the effects of SOD1 mutations, such as gene-silencing approaches, are under investigation [470]. By identifying patients with these and other relevant genetic variations, pharmacogenomics allows for more precise intervention strategies.

#### 9.3.2. Tailoring Therapies Based on Molecular Signatures

Pharmacogenomics also involves analyzing molecular signatures—biomarkers, protein profiles, and gene expression patterns that reveal unique characteristics of a patient’s disease. These signatures provide insight into the specific biological processes active in each case, allowing clinicians to select therapies that directly target the disease mechanisms most relevant to the individual [471].

In Alzheimer’s, for instance, patients with particular biomarker profiles may benefit more from treatments targeting amyloid or tau pathology. Molecular profiling of biomarkers like APOE ε4 status, phosphorylated tau, and neurofilament light chain levels can guide clinicians in selecting amyloid-clearing drugs that patients are likely to respond to [472]. This approach enables the choice of targeted therapies that are better suited to each patient’s distinct disease pathology.

The listed therapies (Table 3) span cutting-edge approaches such as nanotechnology-based drug delivery, stem cell therapy, and pharmacogenomics. Each therapeutic approach targets specific molecular processes within these diseases, with clinical outcomes demonstrating potential benefits and areas for further research.

In ALS, molecular signatures associated with inflammation or oxidative stress help doctors make more informed treatment decisions [483]. Patients with elevated inflammatory markers may benefit from anti-inflammatory therapies, while those showing signs of oxidative damage might respond better to antioxidant interventions [484]. Pharmacogenomics enables the alignment of treatment strategies with the underlying molecular drivers of each patient’s ALS, optimizing treatment effectiveness and potentially slowing disease progression [485].

For HD, pharmacogenomics is particularly valuable in the development of gene-silencing therapies. By examining HTT gene variations, which affect the mutant huntingtin protein production, clinicians can identify patients likely to benefit from antisense oligonucleotides—therapies designed to reduce harmful protein levels [85]. This molecularly tailored approach holds promise for slowing HD progression, targeting the root cause, and offering the potential for long-term impact in genetically compatible patients. Pharmacogenomics is ushering in a new era of precision medicine for neurodegenerative diseases where treatments are chosen based on each patient’s specific genetic and molecular landscape [486]. By aligning therapies with individual disease characteristics, pharmacogenomics enhances treatment effectiveness, minimizes adverse effects, and brings a new level of personalization to neurodegenerative care.

## 10. Conclusions

Molecular research is illuminating new pathways in the treatment of neurodegenerative diseases, leading to a deeper understanding that goes beyond managing symptoms to targeting the fundamental drivers of conditions like Alzheimer’s, Parkinson’s, ALS, and Huntington’s. This review has delved into the intricacies of neurodegeneration, exploring mechanisms from protein misfolding and inflammation to genetic mutations and cellular energy deficits. These insights are not just theoretical; they guide cutting-edge therapeutic approaches aimed at intervening at critical junctures in these diseases.

### 10.1. Integration of Molecular Mechanisms

The complex interplay among the mechanisms driving neurodegeneration is becoming clearer, revealing potential for therapies that address these intertwined pathways simultaneously. Protein misfolding, for example, does more than disrupt cellular function; it also triggers chronic inflammatory responses that exacerbate neuronal damage. Similarly, mitochondrial dysfunction creates oxidative stress that accelerates protein aggregation and impacts energy-dependent cellular processes. Emerging theories suggest that by targeting these interconnected pathways, it may be possible to “reset” the cellular environment, reducing neurotoxicity while promoting resilience.

For instance, there is growing interest in therapies that simultaneously target both protein aggregation and inflammation. Studies on dual-inhibitor compounds—designed to disrupt tau or amyloid-beta accumulation while dampening neuroinflammation—demonstrate early promise in animal models. These approaches reflect a novel shift toward multi-target treatments that consider the cascading effects of protein pathology on cellular inflammation, energy dysregulation, and neuronal survival, aiming for a broader therapeutic impact.

### 10.2. Implications for Future Research

Despite substantial progress, significant questions remain about the exact molecular drivers that set neurodegeneration in motion. Future research is likely to focus on early cellular changes that precede visible disease symptoms, aiming to identify key biomarkers or “molecular signatures” that predict disease onset. Single-cell sequencing and CRISPR screens, now widely adopted, are shedding light on the earliest transcriptional and epigenetic changes in vulnerable neurons, identifying potential points of intervention long before clinical symptoms appear.

One of the most compelling areas of exploration is the influence of the gut–brain axis, which is increasingly linked to neurodegenerative processes. High-citation studies point to gut microbiota as a modulator of systemic inflammation and immune signaling, with microbial imbalance linked to heightened neuroinflammation and even direct impacts on protein misfolding. For example, recent findings suggest that certain bacterial metabolites may cross the blood–brain barrier where they interact with glial cells to influence neuroinflammatory states. Ongoing research examines whether balancing the gut microbiota through diet, probiotics, or prebiotics could serve as an adjunct therapy to reduce neuroinflammatory triggers and support brain health.

Another emerging area is the study of senescent cells in the brain, often referred to as “zombie cells”, which persist in a dysfunctional state, secreting inflammatory signals that impact neighboring neurons. Senolytic therapies, which selectively target and clear these cells, are investigated as a way to reduce chronic inflammation and improve neural health. Early studies show that removing senescent glial cells can improve cognitive function in neurodegenerative animal models, offering a new approach to disease modification.

### 10.3. Translational Potential

The translational potential of these molecular discoveries is enormous, bridging laboratory findings with clinical applications that could fundamentally alter the approach to neurodegenerative diseases. Advanced drug delivery methods, particularly those involving nanotechnology, make it feasible to cross the BBB with greater precision, delivering therapies directly to affected brain regions. Nanoparticles and exosomes are engineered not only to carry drugs but to release them in response to specific cellular conditions, such as oxidative stress or inflammation, which are prevalent in neurodegenerative environments. This type of “smart” delivery could enhance the therapeutic effects of drugs by timing their release with the most active disease processes.

Stem cell and gene therapies are also moving forward with groundbreaking potential. Beyond replacing lost neurons, stem cells are now engineered to release neurotrophic factors or anti-inflammatory molecules, creating a more supportive environment for remaining brain cells. Gene-editing technologies, such as CRISPR-Cas9, are explored for their potential to silence or repair mutant genes directly in the CNS. High-profile trials examine gene therapies that target the root genetic causes in diseases like Huntington’s and familial ALS, offering hope for treatments that go beyond managing symptoms to modifying the underlying disease.

Pharmacogenomics is accelerating this movement toward personalized neurodegenerative therapies, allowing for treatments to be tailored to each patient’s unique genetic makeup. By predicting drug response through genetic testing, pharmacogenomics enables a more individualized approach, minimizing trial-and-error prescribing and reducing adverse effects. Advances in molecular risk modeling further enhance this field, integrating patient data across genomics, proteomics, and clinical history to refine personalized treatment recommendations. Such precision models are poised to revolutionize care by selecting the best treatment for each patient based on a comprehensive view of their molecular profile.

In sum, molecular research is driving a paradigm shift in neurodegenerative treatment, uncovering innovative therapeutic targets and enabling more precise, multi-targeted approaches. By addressing the complex network of interconnected mechanisms that drive these diseases, we are moving closer to developing treatments that not only alleviate symptoms but intervene at key points in the disease process. With continued advances in molecular understanding and targeted technology, there is a real promise that neurodegenerative diseases may become more manageable, and, one day, even preventable.

## Figures and Tables

**Figure 1 ijms-25-12613-f001:**
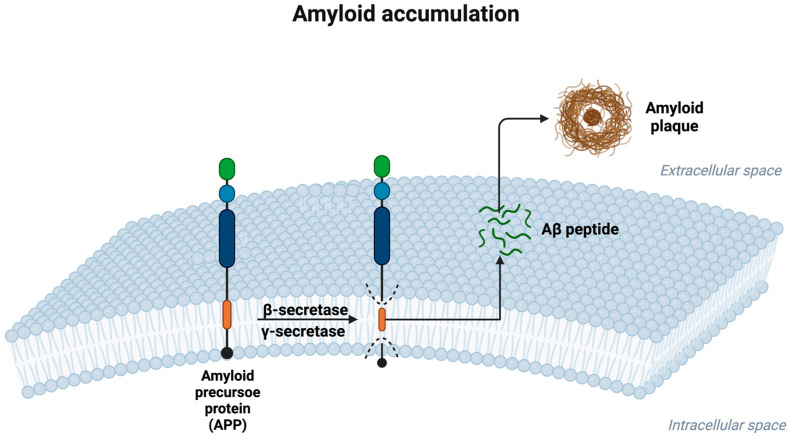
Amyloid accumulation in Alzheimer’s disease. This figure illustrates the process of amyloid-beta (Aβ) peptide formation and accumulation, a hallmark of Alzheimer’s disease pathology. Targeting Aβ oligomers is significant because these smaller aggregates are more neurotoxic than amyloid plaques. Aβ oligomers disrupt synaptic function, impair ion homeostasis, and activate detrimental signaling pathways, making them a critical focus for therapeutic interventions aimed at halting disease progression.

**Figure 2 ijms-25-12613-f002:**
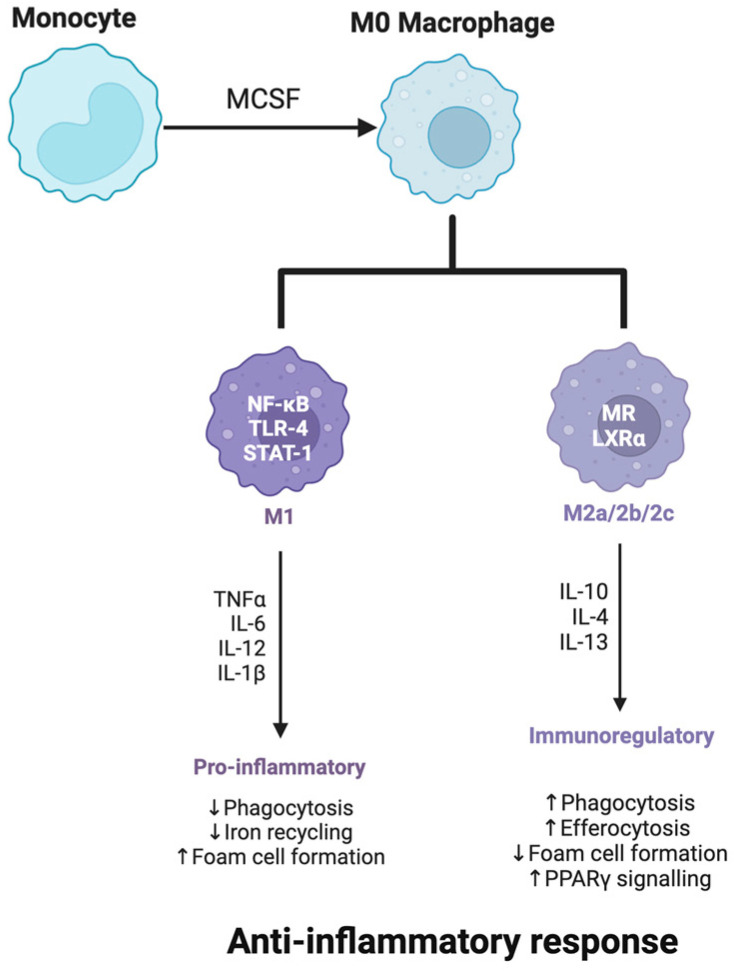
Macrophage polarization and anti-inflammatory response.

**Table 1 ijms-25-12613-t001:** Key molecular mechanisms and their therapeutic targets in neurodegenerative diseases.

Mechanism	Neurodegenerative Disease	Key Molecular Targets	Therapeutic Approaches	Notable Studies (High Citations)
Protein misfolding and aggregation	Alzheimer’s, Parkinson’s	Amyloid-beta, tau, alpha-synuclein	Anti-aggregation agents, tau inhibitors, immunotherapy	Selkoe (2002) [86], Spillantini et al. (1998) [87]
Neuroinflammation	All (AD, PD, ALS, HD)	Microglia, cytokines (TNF-α, IL-1β), NLRP3 inflammasome	Anti-inflammatory agents, cytokine inhibitors	Heneka et al. (2015) [88], Amor et al. (2010) [89]
Mitochondrial dysfunction	ALS, Huntington’s	Electron transport chain complexes, mtDNA	Antioxidants, mitochondrial enhancers	Lin and Beal (2006) [90], Bhatt et al. (2021) [91]
Genetic and epigenetic regulation	Alzheimer’s, ALS, Huntington’s	APOE ε4, SOD1, HTT, DNA methylation, histone modifications	Gene silencing (RNAi, ASOs), CRISPR, epigenetic modulators	De Strooper and Karran (2016) [92], Finkbeiner et al. (2011) [93]
Synaptic dysfunction	Alzheimer’s, Parkinson’s	Cholinergic (ACh), dopaminergic systems, NMDA receptors	Synaptic modulators, neurotrophic factors	Stanciu et al. (2020) [94], Dauer and Przedborski (2003) [95]

**Table 2 ijms-25-12613-t002:** High-citation studies on molecular biomarkers for early detection and diagnosis.

Biomarker Type	Primary Marker(s)	Associated Disease(s)	Detection Method	Diagnostic Value	High-Citation Studies
Protein aggregates	Amyloid-beta, tau	Alzheimer’s	CSF analysis, PET imaging	High sensitivity and specificity for early AD diagnosis	Chapleau et al. (2022) [365], Greenberg et al. (2022) [366]
Inflammatory cytokines	TNF-α, IL-1β, IL-6	ALS, Alzheimer’s, PD	Blood-based biomarkers, CSF sampling	Elevated levels correlating with neuroinflammatory activity	Heneka et al. (2015) [88], Hu et al. (2017) [367]
Genetic mutations	SOD1, HTT, APOE ε4	ALS, Huntington’s, AD	Genotyping, next-gen sequencing	APOE ε4 for AD risk; SOD1 mutations in familial ALS	Serrano-Pozo et al. (2021) [368], Renton et al. (2014) [369]
Oxidative stress markers	8-OHdG, MDA, protein carbonyls	ALS, Huntington’s	Urine, blood assays	Indicative of oxidative damage in early neurodegeneration	Beal et al. (2005) [370], Chen et al. (2004) [371]
Synaptic loss indicators	Synaptophysin, PSD-95	Alzheimer’s, Parkinson’s	Immunohistochemistry, PET tracers	Correlates with cognitive decline and synaptic dysfunction	Tzioras et al. (2023) [372], Dauer et al. (2003) [95]

**Table 3 ijms-25-12613-t003:** Therapeutic innovations and clinical trials in neurodegenerative disease treatments.

Therapeutic Approach	Molecular Target	Disease Focus	Therapeutic Agent(s)	Trial Phase	Key Outcomes	Cited Studies
Nanotechnology-based drug delivery	Blood–brain barrier penetration, tau/amyloid-beta targeting	Alzheimer’s, Parkinson’s	Liposomal encapsulation, PEG-modified nanoparticles	Phase II (various)	Improved brain bioavailability, reduced plaques/tangles in animal models	Hajjo et al. (2022) [473], Saraiva et al. (2016) [474]
Stem cell therapy	Dopaminergic neurons, motor neurons	Parkinson’s, ALS	iPSCs, MSCs	Phase I/II	Motor symptom improvement, neuron survival post-transplant	Takahashi et al. (2007) [475], Glass et al. (2016) [476]
Gene therapy	SOD1, HTT, APOE ε4	ALS, Huntington’s, AD	Antisense oligonucleotides, CRISPR	Phase I/II	Reduced toxic protein levels, slower disease progression	Kordasiewicz et al. (2012) [477], Bennett et al. (2019) [478]
Pharmacogenomics	CYP2D6, COMT, APOE ε4	Alzheimer’s, Parkinson’s	Genotype-guided dosing for existing drugs	Phase II/III	Reduced side effects, improved therapeutic outcomes	Singh et al. (2017) [479], Müller et al. (2015) [480]
Microbiome modulation	Gut–brain axis, inflammation	Parkinson’s, Alzheimer’s	Probiotics, prebiotics, antibiotics	Phase I	Reduced neuroinflammation markers, improved cognitive/motor symptoms	Sampson et al. (2016) [481], Cattaneo et al. (2017) [482]

## Data Availability

Not applicable.

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
