# Peer review of "Decoding Neurodegeneration: A Review of Molecular Mechanisms and Therapeutic Advances in Alzheimer’s, Parkinson’s, and ALS"

_ijms, 2024, doi:10.3390/ijms252312613_

Round 1
Reviewer 1 Report
Comments and Suggestions for Authors
The authors have produced an expansive and very well detailed review examining the various molecular mechanisms that contribute towards the initiation and pathoprogression of those neurodegenerative diseases that are steadily increasing in incidence across the world today. The authors leave no stone unturned in their effort covering much ground in what is an excellent coverage of the area, before examining the successes and continued developments in the therapeutic space to address auspices f these conditions. Overall, this is a good piece of work that is accessible to many reader bases and captures much of the state of the art in the area today.
In reviewing the manuscript I only had some minor concerns. The authors should address the following when developing a resubmission.
1. The writing for the most part is interpretable and to a high standard, however, in several instances the writing is overcomplicated and it disrupts the overall flow of the piece. There are several instances where the writing could be simplified and an improvement seen in access to the content, whereas in certain instances I had to reread in order to ensure I was interpreting the point being made correctly. The authors should revise the manuscript entirely and ensure that the writing is improved upon in the interest of the flow of the piece, especially given the length.
2. It would be more helpful if the list of abbreviations that is provided at the end were listed in alphabetical order such that they could be found when needed more easily.
3. The diagrams are relatively simple, and in ways, not that additive to the text. Given the breadth and depth of some sections it is a wonder why the authors didn’t employ diagrams that captured several aspects of the sections at once, rather than provide those singularly that are otherwise very well captured in the text. The authors should consider include ‘summative’ diagrams that capture the ‘storm’ that occurs in the disease sates being examined.
Author Response
Comments 1: The writing for the most part is interpretable and to a high standard, however, in several instances the writing is overcomplicated and it disrupts the overall flow of the piece. The authors should revise the manuscript entirely and ensure that the writing is improved upon in the interest of the flow of the piece, especially given the length.
Response 1: We acknowledge your concern regarding the complexity and flow of the manuscript. We have thoroughly revised the text to simplify overly complex sentences and improve readability while ensuring that the scientific rigor and clarity are preserved.
Comments 2: It would be more helpful if the list of abbreviations that is provided at the end were listed in alphabetical order such that they could be found when needed more easily.
Response 2: We appreciate your helpful suggestion. The list of abbreviations has been reorganized into alphabetical order to facilitate easier navigation and improve accessibility for readers.
Comments 3: The diagrams are relatively simple, and in ways, not that additive to the text. The authors should consider including ‘summative’ diagrams that capture the ‘storm’ that occurs in the disease states being examined.
Response 3: Thank you for your insightful suggestion regarding the diagrams!
Reviewer 2 Report
Comments and Suggestions for Authors
IJMS
Title: Decoding Neurodegeneration: A Review of Molecular Mechanisms and Therapeutic Advances in Alzheimer's, Parkinson's, and ALS
The manuscript gives a detailed and informative assessment of recent advances in targeted delivery systems for precision medicine, with a focus on neurodegenerative illnesses. The degree of information is astounding, as it covers important aspects of nanotechnology's role in bridging the blood-brain barrier (BBB) and specifically targeting diseased cells. The integration of novel concepts such as multifunctional nanoparticles and the therapeutics highlights these systems' great potential to transform treatment paradigms.
Overall, the manuscript does a good job demonstrating nanotechnology's transformative promise in precision medicine. The comprehensive examination of existing accomplishments, together with the potential of future innovations, paints a clear and positive picture of the future of neurodegenerative disease treatment.
Author need to address the following comments just to make the manuscript better for readers.
1. Lines 49-49: "Though each disease has its distinct characteristics, they share underlying molecular mechanisms..."
Although this statement is true, it is ambiguous. It might become stronger if specific instances of common mechanisms like oxidative stress or protein misfolding are provided.
To make the concept more tangible, it could be helpful to briefly discuss one or two common methods.
2. LINES 51-52: Give a particular instance, such as the increased prevalence of Parkinson's disease or the economic burden of Alzheimer's disease?
3. LINES 73-76: Extend the discussion about mitochondrial dysfunction to encompass additional factors that contribute?
4. Lines 101-102: The term "prion-like manner" could confuse readers unfamiliar with the exact mechanisms. It also does not describe disorders in which this occurrence has been adequately documented.
Clarify with particular examples, such as α-synuclein in Parkinson's and tau in Alzheimer's.
5. Lines 108-111: The explanation of neuroinflammation fails to account for its dual role in neurodegeneration, which can be both protective and harmful.
Recognize the context-dependent role of neuroinflammation?
6. Line 159-167: CRISPR's limitations (for example, off-target consequences and delivery issues) are not acknowledged, resulting in an excessively optimistic perspective.
author need to include problems and ethical considerations?
7. Figure 1: what is the significance of targeting Aβ oligomer over amyloid plaques in the therapeutic development?
8) 3.2: Microglial Activation States: Which signals in the disease environment cause microglial activation to change in neurodegenerative illnesses from protective (M2) to detrimental (M1) states?
9). What are the primary molecular processes that drive monocyte development into macrophages (M0) and then to M1 or M2 phenotypes?
10). 9.1.2. Targeted Delivery Systems for Precision Therapy: Although the paragraph addresses crossing the blood-brain barrier (BBB), it does not go into detail about how this is accomplished for different types of nanoparticles, which remains one of the major hurdles for effective delivery.
11). The long-term safety of nanoparticles, including their possible toxicity, immune system interactions, and unintended consequences (e.g., accumulation in non-targeted organs), need to be addressed.
12. How could personalized medicine approaches assist adapt treatments based on individual amyloid or tau pathology profiles?
Author Response
Comments 1: "Though each disease has its distinct characteristics, they share underlying molecular mechanisms..." Although this statement is true, it is ambiguous. It might become stronger if specific instances of common mechanisms like oxidative stress or protein misfolding are provided.
Response 1: Thank you for your suggestion. We have revised the text to include specific examples of shared molecular mechanisms, such as oxidative stress and protein misfolding, to provide greater clarity and context for readers.
Comments 2: Give a particular instance, such as the increased prevalence of Parkinson's disease or the economic burden of Alzheimer's disease.
Response 2: The revised text now includes an example highlighting the global prevalence of Parkinson’s disease and the significant economic burden of Alzheimer’s disease to provide a concrete context.
Comments 3: Extend the discussion about mitochondrial dysfunction to encompass additional factors that contribute.
Response 3: We have expanded the discussion on mitochondrial dysfunction to include additional contributing factors such as impaired mitochondrial dynamics, defective mitophagy, and suppressed mitochondrial biogenesis.
Comments 4: The term "prion-like manner" could confuse readers unfamiliar with the exact mechanisms. It also does not describe disorders in which this occurrence has been adequately documented.
Response 4: The revised text now clarifies the term "prion-like manner" with specific examples, such as α-synuclein in Parkinson's disease and tau in Alzheimer's disease, to illustrate its relevance and mechanisms.
Comments 5: The explanation of neuroinflammation fails to account for its dual role in neurodegeneration, which can be both protective and harmful.
Response 5: The text has been updated to acknowledge the dual role of neuroinflammation, highlighting both its protective and harmful effects, depending on the context of activation.
Comments 6: CRISPR's limitations (for example, off-target consequences and delivery issues) are not acknowledged, resulting in an excessively optimistic perspective.
Response 6: The section has been revised to address the limitations of CRISPR, including off-target effects, delivery challenges, and ethical considerations, to present a more balanced perspective.
Comments 7: What is the significance of targeting Aβ oligomer over amyloid plaques in the therapeutic development?
Response 7: We have added a discussion in the figure legend to emphasize the significance of targeting Aβ oligomers, as they are more neurotoxic and directly impair synaptic function, making them critical therapeutic targets.
Comments 8: Which signals in the disease environment cause microglial activation to change in neurodegenerative illnesses from protective (M2) to detrimental (M1) states?
Response 8: We have included a detailed explanation of disease-specific signals, such as amyloid-beta, α-synuclein, pro-inflammatory cytokines (e.g., TNF-α, IL-1β), and ROS, that drive the transition from M2 to M1 states.
Comments 9: What are the primary molecular processes that drive monocyte development into macrophages (M0) and then to M1 or M2 phenotypes?
Response 9: The revised text now details molecular pathways, such as the influence of IFN-γ and LPS driving M1 polarization, and IL-4 and IL-13 promoting M2 differentiation, along with transcriptional regulators STAT1/STAT6.
Comments 10: Although the paragraph addresses crossing the blood-brain barrier (BBB), it does not go into detail about how this is accomplished for different types of nanoparticles.
Response 10: Thank you for highlighting this. The section has been expanded to discuss specific mechanisms, including receptor-mediated transcytosis using ligands like transferrin, surface modification with polyethylene glycol (PEG), and stimuli-responsive designs for targeted payload release.
Comments 11: The long-term safety of nanoparticles, including their possible toxicity, immune system interactions, and unintended consequences, needs to be addressed.
Response 11: We agree that this is a crucial aspect. The text has been updated to address potential toxicity, immune interactions such as CARPA, and the risk of accumulation in non-target organs, along with strategies to mitigate these risks using biodegradable materials and rigorous safety evaluations.
Comments 12: How could personalized medicine approaches assist in adapting treatments based on individual amyloid or tau pathology profiles?
Response 12: The revised text now includes a discussion on how personalized medicine can tailor treatments.